# South Asia anthropogenic ammonia emission inversion through assimilating IASI observations

Ji Xia[1,*], Yi Zhou[2,*], Li Fang[1], Yingfei Qi[3], Dehao Li[1], Hong Liao[1], and Jianbing Jin[1]

[1]State Key Laboratory of Climate System Prediction and Risk Management, Jiangsu Key Laboratory of Atmospheric Environment Monitoring and Pollution Control, Jiangsu Collaborative Innovation Center of Atmospheric Environment and Equipment Technology, School of Environmental Science and Engineering, Nanjing University of Information Science and Technology, Nanjing, Jiangsu, China
[2]School of Management Science and Engineering, Nanjing University of Information Science and Technology, Nanjing, Jiangsu, China
[3]College of Geography and Remote Sensing, Hohai University, Nanjing, Jiangsu, China
[*]These authors contributed equally to this work.

**Correspondence:** Jianbing Jin (jianbing.jin@nuist.edu.cn)

**Abstract.** Ammonia has attracted significant attention due to its pivotal role in the ecosystem and its contribution to the formation of secondary aerosols. Developing an accurate ammonia emission inventory is crucial for simulating atmospheric ammonia levels and quantifying its impacts. However, current inventories are typically constructed in the bottom-up approach and are associated with substantial uncertainties. To address this issue, assimilating observations from satellite instruments for top-down emission inversion has emerged as an effective strategy for optimizing emission inventories. Despite the severity of ammonia pollution in South Asia, research in this context remains very limited. This study aims to estimate ammonia emissions in this region by integrating the prior emission inventory from the Community Emissions Data System (CEDS) and the columned ammonia concentration retrievals from the Infrared Atmospheric Sounder Interferometer (IASI). We employ a newly-developed four-dimensional ensemble variational (4DEnVar)-based emission inversion system to conduct the calculations, resulting in monthly ammonia emissions for 2019 at a resolution of $0.5° \times 0.625°$. The annual total estimate for the posterior emission inventory is 12.61 Tg, compared to the prior inventory's 13.32 Tg. Our simulations, driven by the posterior emission inventory, demonstrate superior performance compared to those driven by the prior emission inventory. This is validated through comparisons against the IASI observations, the independent column concentration measurements from the advanced satellite instrument Crosstrack Infrared Sounder (CrIS), and the ground concentration observations of ammonia and $PM_{2.5}$. Additionally, the spatial and temporal characteristics of ammonia emissions in South Asia based on the posterior are analyzed. Notably, emissions there exhibit a "double-peak" seasonal profile, with the maximum in July and the secondary peak in May. This differs from the "double-peak" trend suggested by the CEDS prior inventory, which identifies the maximum column concentration in May and a second peak in September. The differences may be attributed to a more accurate representation of regional agricultural practices, such as the timing of fertilizer application and meteorological influences like precipitation and temperature.

# 1    Introduction

Ammonia ($NH_3$), an alkaline compound, has the capacity to react with acidic gases present in the atmosphere, thereby contributing to the formation of secondary aerosols, notably ammonium sulfate and ammonium nitrate (Jimenez et al., 2009). The genesis of fine atmospheric particulate matter poses significant threats to human health (Mukherjee and Agrawal, 2017). Further, ammonia gas, along with its reaction products, plays a pivotal role in soil acidification and the eutrophication of water bodies through both dry and wet deposition (Krupa, 2003), and thereby affecting the balance of ecosystems (Asman et al., 1998) and climate change (Ma et al., 2022; Gong et al., 2024). With an enormous livestock population and extensive use of nitrogen fertilizers, South Asia has experienced the highest level of atmospheric $NH_3$ globally (Pawar et al., 2021b; Luo et al., 2022). Specifically, the annual average ammonia concentration across India is approximately $1.8–5.6 \times 10^{16}$ mol cm$^{-2}$, while in the Indo-Gangetic Plain (IGP) of India, the concentration is double that of other regions, reaching a peak of $11.5 \times 10^{16}$ mol cm$^{-2}$ during the high season in July (Kuttippurath et al., 2020).

Over the past decade, scientists have predominantly employed the "bottom-up" approach to estimate $NH_3$ emissions. When combined with chemical transport models, atmospheric $NH_3$ dynamics can be simulated, enabling the quantification of environmental impacts. Substantial efforts have been made to quantify the spatiotemporal distribution of $NH_3$ sources and develop global/regional emission inventories, such as the global $NH_3$ emission inventory (Bouwman et al., 1997), the anthropogenic emission inventory that includes $NH_3$ estimates (e.g., Community Emissions Data System, CEDS) (Hoesly et al., 2018), as well as regional $NH_3$ inventories focusing on South Asia (Yan et al., 2003; Yamaji et al., 2004; Liu et al., 2022). However, these bottom-up estimates of $NH_3$ emissions are generally considered as uncertain (Xu et al., 2019), particularly when compared emissions of other pollutants primarily originating from fossil fuel combustion such as $NO_2$. One challenge is that the intensity of agricultural $NH_3$ emissions (i.e., emission factors), whether from livestock or fertilizer, depends heavily on management and farming practices, but this information is often not readily available (Zhang et al., 2017). As a result, atmospheric chemistry transport models driven by these emission estimates inevitably struggle to reproduce atmospheric $NH_3$ concentrations. Consequently, these discrepancies hinder our comprehensive understanding of the environmental implications of $NH_3$ emissions.

The rapid advancement of satellite remote sensing technology has resulted in an expanding array of valuable $NH_3$ products, such as those from the first satellite $NH_3$ observations using the Tropospheric Emission Spectrometer (TES) (Beer et al., 2008), as well as higher-resolution retrievals from the Infrared Atmospheric Sounding Interferometer (IASI) (Pawar et al., 2021b) and the Cross-track Infrared Sounder (CrIS) (Beale et al., 2022; Kharol et al., 2022). While these remote sensing measurements play a pivotal role in characterizing atmospheric $NH_3$ loading, limitations still remain. These primarily arise from the fact that satellite observations can only measure column-integrated $NH_3$ concentrations, which do not directly reflect emission intensity or the three-dimensional concentration field. In addition to these satellite-based data, very limited ground-based observations are publicly available over South Asia, and those that do exist are constrained by their inadequate representation of atmospheric $NH_3$ features (Pawar et al., 2021b).

In the field of atmospheric pollutant modeling, an alternative method for calculating emission flux is the "top-down" approach, which is achieved through data assimilation. Over the past decade, emission inversion has gained widespread attention globally and has been applied in various contexts, including the estimation of Volatile Organic Compounds (VOCs) (Bauwens et al., 2016; Choi et al., 2022), sulfur dioxide ($SO_2$) (Qu et al., 2019; Li et al., 2021), methane ($CH_4$) (Wecht et al., 2014; Fujita et al., 2020), and atmospheric $NH_3$ emissions. For example, Kong et al. (2019) calculated the 2016 $NH_3$ emission inventory in China by assimilating ground-based $NH_3$ concentration observations from several dozen ground stations. Similarly, Chen et al. (2021) optimized the prior $NH_3$ emission estimates from the United States' 2011 National Emissions Inventory (2011 NEI) by assimilating $NH_3$ column concentrations from IASI instruments across the United States. Recently, we developed a four-dimensional variational assimilation-based $NH_3$ emission inversion system, which has been successfully tested in $NH_3$ emission inversion by assimilating IASI products over China.

However, there is a paucity of studies focusing on assimilation-based $NH_3$ emission inversion specific to South Asia, which has some of the highest $NH_3$ loading hotspots compared to other continents. In this study, we aim to explore the spatial and temporal features of $NH_3$ emissions over South Asia. The $NH_3$ emission inventory will be calculated using our newly developed emission inversion system (Jin et al., 2023), by assimilating $NH_3$ retrievals from the IASI instruments onboard MetOp-A (operational from 2008 to 2018), MetOp-B (operational since 2012), and MetOp-C (operational since 2018) satellites. Instead of directly assimilating IASI measurements as previous studies have done, we incorporated the averaging kernel information from the latest version of the IASI product. This approach allowed us to update the column concentration observations before assimilation. By doing so, we ensure a fairer comparison between the simulated and observed columnar $NH_3$ concentrations, a point that has been emphasized in several studies (Eskes and Boersma, 2003; von Clarmann and Glatthor, 2019), but never implemented in the IASI-based emission inversion. We aim to provide a more accurate estimation of anthropogenic $NH_3$ emission inventories and to explore their spatial and temporal characteristics across South Asia. Additionally, it serves as a model for effectively calculating atmospheric pollution emissions in regions that have been less studied in the past. The study focuses on anthropogenic $NH_3$ emissions but also contributes to a broader understanding of atmospheric pollution in under-researched regions.

The remaining sections of this paper are organized as follows: Section 2 describes the measurements assimilated in the $NH_3$ emission inversion, as well as those used for independent validation. The assimilation methodology for the emission inversion, along with the choice of the prior emission inventory and the chemical transport model, is also outlined. Section 3 presents the validation results of the emission inversion and highlights the key features of $NH_3$ emissions over South Asia.

## 2 Data and method

### 2.1 IASI satellite measurements

IASI (Infrared Atmospheric Sounding Interferometer) is a Fourier Transform Spectrometer that operates in the thermal infrared spectral range. It is onboard the Meteorological Operational (MetOp) A/B/C satellites, a series of European polar-orbiting meteorological satellites managed by the European Space Agency (ESA) and the European Organization for the

Exploitation of Meteorological Satellites (EUMETSAT). The first MetOp-A satellite, equipped with IASI, was launched in 2008, followed by MetOp-B and MetOp-C in 2012 and 2018, respectively. The IASI instruments operate at an altitude of 817 km in a sun-synchronous orbit with an inclination of 98.7 °. Each instrument conducts measurements over a ground swath width of 2200 km, with 30 fields of view (15 on each side of the nadir). Each field of view consists of four pixels, each with a nadir diameter of 12 km. This observational strategy enables each IASI instrument to make two passes over every point on Earth daily, around 09:30 and 21:30 local time (Bouillon et al., 2020).

The assimilated observations for estimating the $NH_3$ emissions were the monthly IASI column concentration means over the 0.5 °× 0.625 °GEOS-Chem grid cell. These values were derived from the latest ANNI-$NH_3$-v4R-ERA5 product. Despite improvements in $NH_3$ column retrievals from satellite observations, there remains substantial variability in measurement uncertainty, ranging from 5% to over 1000% (Van Damme et al., 2014; Whitburn et al., 2016; Van Damme et al., 2017). Data selection was performed by excluding nighttime observations, irrational values (<0), and only using data with a cloud fraction < 0.1 (Van Damme et al., 2018) and skin temperature > 263 K (Van Damme et al., 2014) during the calculation of the monthly mean. While negative values are not necessarily incorrect, they are considered unrealistic in the context of $NH_3$ concentrations. To improve the quality of the monthly average, we removed those negative values. Additionally, we have re-compared the cases of excluding negative $NH_3$ total column values versus retaining them. As shown in Fig. S1, the positive bias on the final concentrations within our study region is minimal. It is also important to note that we used daily observations from three satellites, each with a pixel resolution of approximately 12 km × 12 km, which provided us with sufficient observations to calculate the monthly average. We applied a selection criterion, using only grid averages that contain a minimum of 80 observations. This ensures that the grid-averaged values are statistically representative and that the monthly mean is of high quality. Notably, the time coverage of the available version 4 IASI product used was limited: Metop-A provided data for the entire year of 2019, Metop-B provided data from January to July 2019, and Metop-C did not have data for 2019. Therefore, only the data from Metop-A and Metop-B within the 2019 time frame were used in this study. The use of both Metop-A and Metop-B data for 2019 ensures data continuity and enhances the reliability of the measurements. While a single satellite could provide sufficient data, using both platforms could improve temporal and spatial coverage, resulting in more accurate and robust results. To ensure robustness, we have also made a brief comparison of the $NH_3$ column concentrations obtained from both Metop-A and Metop-B satellites, as shown in the Fig. S2. Despite some small differences, the data from both satellites are generally consistent in terms of spatial patterns and concentration levels. Additionally, the data from the two satellites can complement each other, indicating good reliability of the results across both platforms. To further improve the data quality and ensure consistency, we performed monthly and grid averaging of the observations. This approach not only allows for a fair comparison between the observed and modeled $NH_3$ concentrations but also reduces the computational cost of the assimilation process. Using individual observations without averaging would result in an excessively large observational vector, which would significantly increase the computational burden. For example, without averaging, the size of the observational vector could reach 1,000,000, while with monthly and grid averaging, it is reduced to a manageable size of around 1,000. This reduction in size helps to optimize the data assimilation process while maintaining the integrity of the emission estimates.

Compared to the previous version, one highlight of the lastest version 4 product is that it includes averaging kernel information. The benefit of using the averaging kernel is that it can consider the vertical distribution characteristics of satellite observations, helping to correct the satellite retrieval results and making them more representative of the true distribution of the target gas or variable in the atmosphere (Rodgers, 2000). The impact of averaging kernels (AVKs) are supposed to be considered in the data processing. The sensitivity of IASI $NH_3$ observations varies with altitude, and AVKs enable the adjustment of simulated or observed $NH_3$ concentrations to align with the vertical distribution detected by IASI. This adjustment is particularly important for data comparison and validation against the model simulations (Clarisse et al., 2023). The formula for calculating the column concentration, after accounting for the averaging kernels, in this paper follows:

$$\hat{X}^m = \frac{\hat{X}^a - B}{\sum_z A_z^a m_z} + B. \tag{1}$$

here, $\hat{X}^m$ represents the IASI column concentration retrieved with model profile. $\hat{X}^a$ denotes the initial IASI column concentration, with the background concentration B. The $A_z^a$ values are AVK for each vertical layer, with the model profile $m_z$. More detailed information and the corresponding equations are provided in the supplementary materials equation S8 and S9.

The uncertainty assigned to the IASI measurements is also an essential. When calculating the uncertainty of gridded monthly average $NH_3$ measurements, both instrumental errors $\sigma^{\text{instrumental}}$ and representation error $\sigma^{\text{representation}}$ are considered. The gridded average uncertainty derived directly from IASI products was designated as instrumental error $\sigma^{\text{instrumental}}$, while the standard deviation of the observed samples for the gridded average characterized representation error $\sigma^{\text{representation}}$. The total uncertainty $\sigma^{\text{integrated}}$ for weighting the potential spread of the assimilated IASI $NH_3$ measurements is finally expressed as:

$$\sigma^{\text{integrated}} = \left\{ \left( \sigma^{\text{instrument}} \right)^2 + \left( \sigma^{\text{representing}} \right)^2 \right\}^{0.5} \tag{2}$$

Four snapshots of the assimilated monthly IASI $NH_3$ column concentration observations and their uncertainty in January, April, July and November can be found in Fig. 1 (a) and Fig. S3. These four scenarios are selected to highlight the typical seasonal profile of the $NH_3$ loading over South Asia.

## 2.2    Independent observations for validation

The Crosstrack Infrared Sounder (CrIS) $NH_3$ column concentration and ground-based observations of $NH_3$ and $PM_{2.5}$ from the Central Pollution Control Board (CPCB) of India were also collected to validate our assimilation results.

The CrIS instrument was launched in 2011 on the Suomi National Polar-Orbiting Partnership (SNPP) satellite and in 2017 on the NOAA-20 satellite. The retrieval products from SNPP began in 2011 and ended in May 2021, with a data gap from April to August 2019. The $NH_3$ retrieval products from NOAA-20 started in March 2019. Therefore, we used retrieval products from both SNPP and NOAA-20 as independent observations for 2019. We utilized the Level 2 CrIS product from the CFPR 1.6.4 version. Specifically, only the CrIS observations during daytime, under cloud-free conditions, and with a quality flag $\geq 3$ were selected. These original data were subsequently interpolated to achieve a spatial resolution of 0.5 $°\times$ 0.625 $°$, which is consistent with our $NH_3$ simulation. Similarly, we also considered the impact of the averaging kernels (AVKs) and

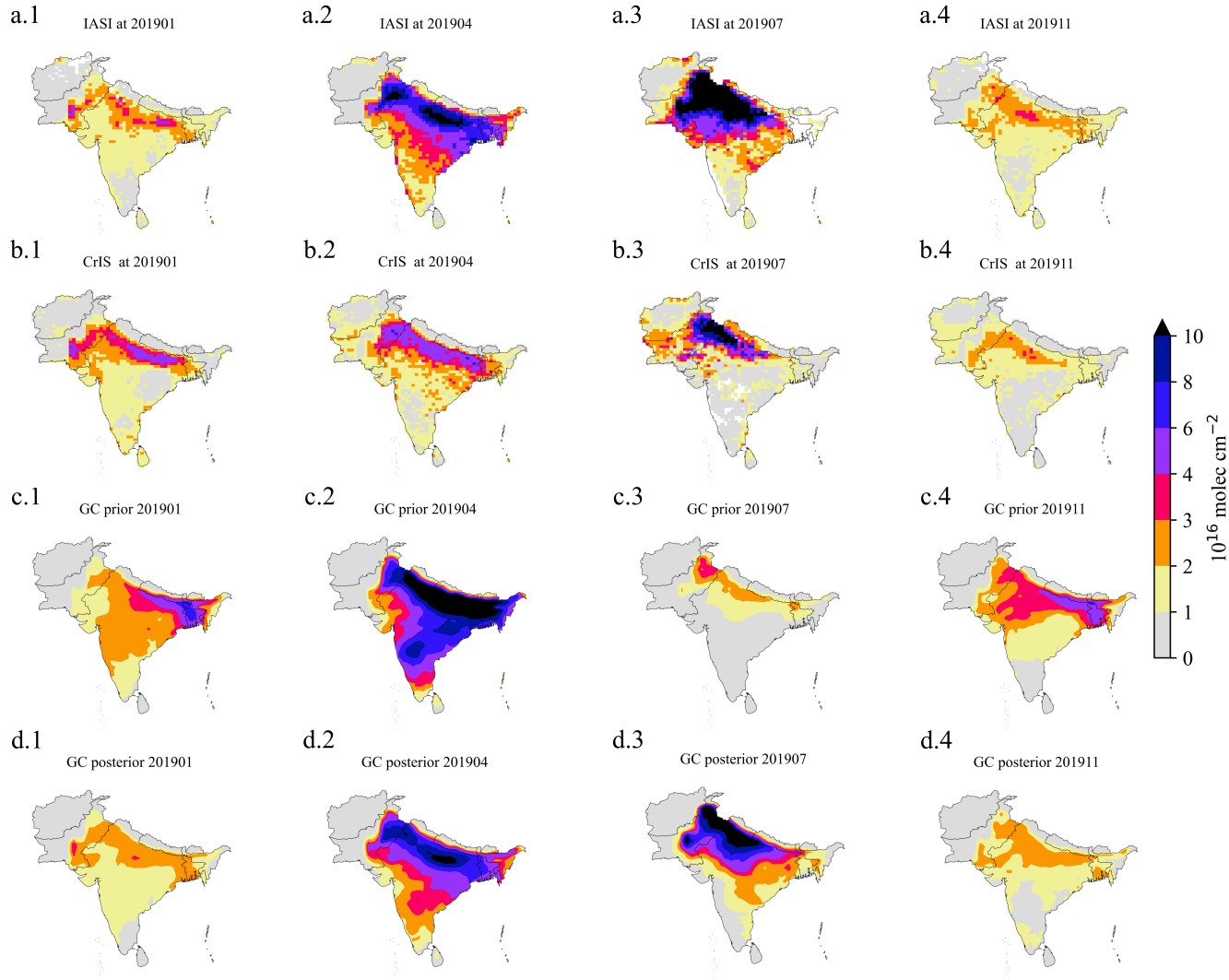

**Figure 1.** Spatial distribution of the total column NH$_3$ concentration from IASI (a) or CrIS (b) instruments, and from the GEOS-Chem simulation either using the prior (c) or using the posterior (d) NH$_3$ emission flex in 2019 January (a.1)–(d.1), April (a.2)–(d.2) , July (a.3)-(d.3) and November (a.4)–(d.4).

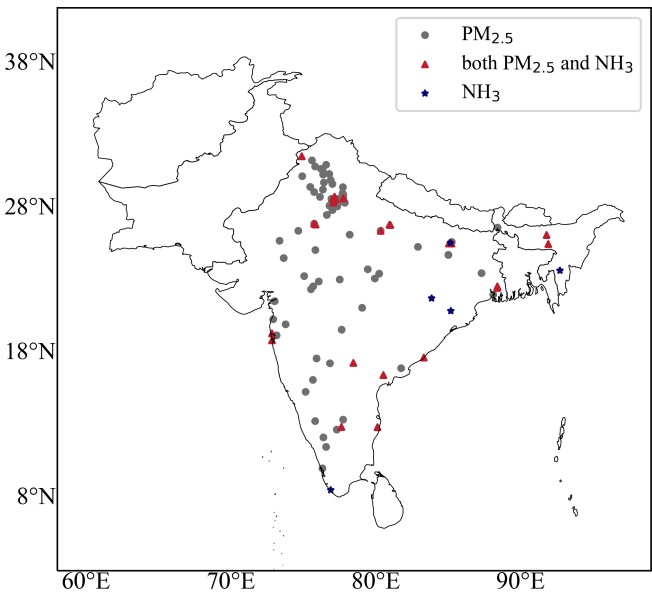

**Figure 2.** The GEOS-Chem model simulation domain, with dots indicating the locations of ground observation stations from the Central Pollution Control Board (CPCB), India. The three different colored dots represent stations with only $PM_{2.5}$ observations, stations with both $PM_{2.5}$ and $NH_3$ observations, and stations with only $NH_3$ observations, respectively.

applied the AVKs to the satellite profile data. We converted the logarithmic averaging kernels into linearized averaging kernels based on the method proposed by Cao et al. (2022).

Ground observations of $NH_3$ in South Asia are mainly provided by the Central Pollution Control Board (CPCB, https://cpcb.nic.in/), which is the official portal of Government of India. $NH_3$ is measured by the chemiluminescence method as $NO_x$ following the oxidation of $NH_3$ to $NO_x$. In that approach, $NH_3$ is determined from the difference between NOx concentration with and without inclusion of $NH_3$ oxidation (Pawar et al., 2021b). The ground level $NH_3$ concentration data from CPCB was successfully collected. There were $NH_3$ surface concentration observations from 33 stations available in 2019, and the distribution are shown in Fig. 2.

$PM_{2.5}$ observations from CPCB were also used in the assimilation validation. The $PM_{2.5}$ observations were selected before they were used, which follows (Spandana et al., 2021): First, select the hourly $PM_{2.5}$ data greater than $PM_{10}$, then filter out the hourly $PM_{2.5}$ data that falls outside the range of $day_{mean}$ - 3 × standard deviation and $day_{mean}$ + 3 × standard deviation. Additionally, ensure that each day contains at least 20 hours of data. Finally, the data is processed into monthly averages for subsequent validation. The distribution of the ground stations where the $PM_{2.5}$ were used in this paper can be found in Fig. 2 and detailed information about the stations is also provided in Table S1-S3.

## 2.3 Emission inversion system

This study employs the four-dimensional ensemble variational (4DEnVar) data assimilation -based NH₃ emission inversion system that was developed by Jin et al. (2023). The general idea of assimilation-based emission inversion is to find the most likely estimate, which in this case is the monthly NH₃ emission field, given the prior NH₃ emissions and the observations. The calculation is conducted through minimizing the cost function $\mathcal{J}$:

$$\mathcal{J}(\boldsymbol{f}) = \frac{1}{2}(\boldsymbol{f} - \boldsymbol{f}_b)^{\mathrm{T}} \mathbf{B}^{-1} (\boldsymbol{f} - \boldsymbol{f}_b) + \frac{1}{2}\{\boldsymbol{y} - \mathbf{H}\mathcal{M}(\boldsymbol{f})\}^{\mathrm{T}} \mathbf{O}^{-1}\{\boldsymbol{y} - \mathbf{H}\mathcal{M}(\boldsymbol{f})\} \tag{3}$$

Here, $\boldsymbol{f}$ denotes the vector of the NH₃ estimated emission field, with its units typically expressed in kg/m²/s. Additionally, $\boldsymbol{f}_b$ denotes the prior monthly NH₃ emission vector from CEDS as will be described in Section 2.4. $\mathbf{B}$ represents the background error covariance matrix associated with the prior emission estimate. Here we assumed that the uncertainty in the NH₃ emission can be compensated by a spatially varying tuning factor $\alpha$. The $\alpha$ values are defined to be random variables with a mean of 1.0 and a standard deviation $\sigma = 0.2$. In addition, a correlation matrix $\mathbf{C}$ is introduced for quantifying the spatial correlation between two $\alpha$s in the grid $i$ and $j$ as:

$$\mathbf{C}(i,j) = e^{-(d_{i,j}/l)^2/2} \tag{4}$$

where $d_{i,j}$ represents the distance between two grid cells $i$ and $j$. $l$ here denotes the correlation length scale which controls the spatially variability freedom of the $\alpha$s. An empirical parameter $l = 300$ km which is used in the NH₃ emission inversion in China (Jin et al., 2023) is also used in this study. With the spatial correlation matrix and the emission uncertainty, the background error covariance matrix could then be constructed as:

$$\mathbf{B}(i,j) = \sigma^2 \cdot \boldsymbol{f_b}(i) \cdot \boldsymbol{f_b}(j) \cdot \mathbf{C}(i,j) \tag{5}$$

$\mathcal{M}$ here represents the GEOS-Chem model (as will be illustrated in Section 2.4) driven by the emission $\boldsymbol{f}$, $\mathbf{H}$ here is the observational operator that transfer the simulated NH₃ 3D concentration result into the observational space. $\boldsymbol{y}$ represents the monthly IASI NH₃ column concentration observations, while $\mathbf{O}$ is the observation error covariance matrix. Here we assume IASI observation representation errors are independent from each other. $\mathbf{O}$ therefore is a diagonal matrix filled with the square of the integrated uncertainty as described in Section 2.1. Meanwhile, a minimum measurement error is used to prevent the posterior from being too close to low-value observations, thereby avoiding model divergence:

$$\mathbf{O}_{i,i} = \min\left(1.0 \times 10^{16} \text{ molec cm}^{-2}, \sigma^{\text{integrated}}\right)^2 \tag{6}$$

More information about how we minimizing the cost function Eq. 3 could be fund in Jin et al. (2023).

## 2.4 GEOS-Chem model and emission inventory

GEOS-Chem, a three-dimensional (3-D) global tropospheric chemistry model, is driven by assimilated meteorological data obtained from the Goddard Earth Observing System (GEOS) at the NASA Data Assimilation Office (DAO) (Bey et al.,

| Data and Model | Period | Use |
|---|---|---|
| IASI v3 | 2015-2023 | Annual variation of $NH_3$ concentration |
| IASI v4 | entire 2019 | Inversion and Validation |
| Level 2 CrIS | entire 2019 | Independent validation |
| CPCB | entire 2019 | Independent validation |
| GEOS-Chem | entire 2019 | Similation |

**Table 1.** The use of observations and simulations

2001). GEOS-Chem incorporates a fully integrated chemistry system involving aerosol, ozone, $NO_x$, and hydrocarbons, as described by Park et al. (2004).The wet deposition scheme is explained by Liu et al. (2001) for water-soluble aerosols and by Amos et al. (2012) for gaseous components. Dry deposition is modeled using the resistance-in-series scheme proposed by Wesely and Lesht (1989), as applied by Wang and Jacob (1998). Size-specific aerosol dry deposition follows the approach
outlined by Emerson et al. (2020). A nested grid simulation within the GEOS-Chem model v13.4.1 is conducted to simulate the atmospheric environment over South Asia. The nested domain (60 °–98 °E, 4 °–40 °N), shown in Fig. 2, has a horizontal resolution of 0.5 °latitude by 0.625 °longitude, accompanied by 47 vertical layers. The model is driven by meteorological fields from the Modern-Era Retrospective analysis for Research and Applications, Version 2 (MERRA-2) reanalysis dataset provided by the Global Modeling and Assimilation Office (GMAO) at NASA. The model employs a three-month spin-up
period to minimize the influence of initial conditions. Lateral boundary conditions for the nested domain are updated every 3 hours using output from the global GEOS-Chem simulation at 2 °× 2.5 °resolution. Chemical initial conditions are also obtained from the global simulation to ensure consistency.

     The $NH_3$ emissions inventory employed to drive GEOS-Chem originated from the Community Emissions Data System (CEDS, https://doi.org/10.25584/PNNLDH/1854347) inventory, which was widely used for modeling the South Asia atmo-
spheric pollutants, e.g., VOCs (Chaliyakunnel et al., 2019), $PM_{2.5}$ pollution (Guttikunda and Nishadh, 2022; Xue et al., 2021). CEDS inventory includes various sources encompassing agricultural, energy production, industrial, residential and commercial activities, ships, solvent use, surface transportation, and waste processing (McDuffie et al., 2020), the bulk of $NH_3$ emissions originate from agricultural practices. Specifically, these emissions stem predominantly from farmlands, including crops such as wheat, maize, and rice, as well as manure from livestock, including cattle, chicken, goats, and pigs (Liu et al., 2022). The CEDS
emission estimates were coarse-grained into the model resolution $0.5° × 0.625°$ before being utilized to drive the GEOS-Chem simulations. Examples of the CEDS emission over the South Asia are presented in Fig. 3, which plot the total $NH_3$ emission fluxes for January, April, July, November of the year 2019. Additionally, the model's biogenic emissions are based on the MEGAN2.1 (Model of Emissions of Gases and Aerosols from Nature) inventory (Guenther et al., 2012), while the biomass burning sources driving the model are based on the GFEDv4 (Version 4 of the Global Fire Emissions Database) inventory
(Giglio et al., 2013). The use of IASI and CrIS observations, along with GEOS-Chem simulations, is outlined in Table 1.

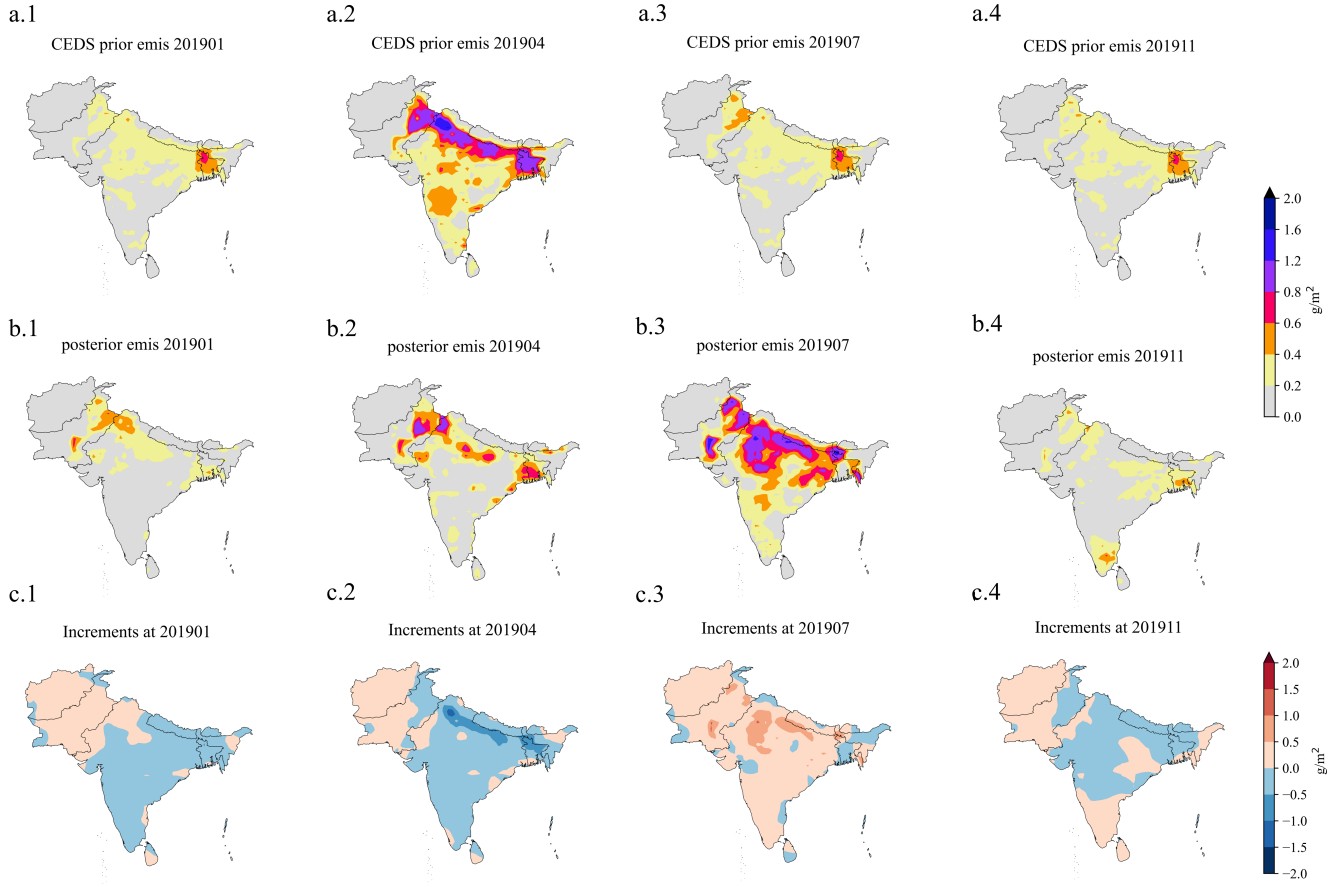

**Figure 3.** Spatial distribution of the prior (a), the posterior (b) and the posterior minus prior increments (c) monthly NH$_3$ emission in 2019 January (a.1)–(c.1), April (a.2)–(c.2) , July (a.3)-(c.3) and November (a.4)–(c.4).

## 3 Results and discussion

With the assimilation system described above, the monthly anthropogenic NH$_3$ emission inversion for 2019 over South Asia is conducted. The Spatial of prior and posterior results are in Section 3.1.1. The long-term varying trend of South Asia NH$_3$ emission is illustrated in Section 3.1.2, followed by an analysis and discussion of its spatial distribution and seasonal profile based on the inversion results in Section 3.2. Then the posterior result is evaluated in Section 3.3.

### 3.1 Observed NH$_3$ concentrations

We first present the spatial distribution of NH$_3$ column concentrations from satellite observations and model results driven by either the prior or posterior inventories. Then, we examine their seasonal variation in 2019 and the long-term trends from 2015 to 2023.

### 3.1.1 Spatial NH$_3$ total column concentration

The prior and posterior snapshot of NH$_3$ column concentration simulations for four months (January, April, July, and November) are presented in Fig. 1 (c)-(d), alongside the IASI measurements shown in panel (a). These months were selected as typical examples representing four different seasons. The column concentration distributions for the rest months from the model and satellite observations could be found in Fig. S4 and S5, respectively. While the prior simulation generally captured the distribution of NH$_3$, with hot spots in North India, Pakistan, and Bengal—consistent with the IASI retrievals—it failed to capture the correct seasonal profile. According to the IASI measurements, NH$_3$ concentrations peak in July, a pattern clearly visible in the monthly variation of the IASI-observed NH$_3$ column concentrations from 2015 to 2023 as will be discussed in Section 3.1.2. However, the prior model incorrectly indicated that the highest NH$_3$ loading occurred in the Spring and Autumn seasons. As a result, NH$_3$ loading was severely overestimated in Winter and Spring (particularly in May) but significantly underestimated in Summer.

Note that there are still some discrepancies in the posterior simulation vs IASI column measurements. In particular, as shown in panel a.3 vs. d.3 of Fig. 1, the posterior simulation did not fully reproduce the extremely high NH$_3$ loading observed by IASI in July (with column-integrated concentrations exceeding $10 \times 10^{16}$ molec cm$^{-2}$). This occurs because the goal of the assimilation is to achieve the best fit between the posterior, the observations, and the prior emissions, rather than just fitting the observations alone. The extremely high NH$_3$ concentrations are less likely given the relatively low prior NH$_3$ emissions and the background error covariance matrix described in Section 2.3. Additionally, the 4DEnVar assimilation algorithm inherently accounts for potential model variations through ensemble simulations. However, the response of GEOS-Chem NH$_3$ simulations to emission variations is nonlinear, making it difficult to accurately resolve these discrepancies through the 4DEnVar data assimilation algorithm without implementing outer-loop optimization strategies. Additionally, the spatial distribution of the NH$_3$ column concentrations observed by CrIS, as shown in panel (b) of Fig. 1, demonstrate good consistency with both the IASI observations and the posterior simulation results presented in Fig. 1.

### 3.1.2 Seasonal and annual variation of NH$_3$ concentration

We examined the monthly average of the total NH$_3$ column concentrations simulated by the model over the South Asia region, along with IASI and CrIS observations, in Fig. 4 (a). The prior model results demonstrate significant seasonal variability in NH$_3$ column concentrations, characterized by peaks in May and September and comparatively low levels during the summer months. This has been corrected through assimilating the IASI measurements in this study. Conversely, the posterior results reveal a distinct temporal pattern, featuring a pronounced peak in May and a negligible peak in July. The high value in May is attributed to huge amount of biomass burning in South Asia during the spring in Fig. S6 (c). We have identified the planting and harvesting times of crops in the South Asia region from USDA(U.S.DEPARTMENT OF ARGRICULTURE, https://ipad. fas.usda.gov/rssiws/al/crop_calendar/sasia.aspx), as presented in Table 2. The heavy use of fertilizers in agricultural activities has resulted in the highest emission throughout the year, as will be illustrated in Fig. 4 (b) in Section 3.2. This has lead to the second NH$_3$ concentration peak in July. The reasons for higher emissions in July but lower concentration levels compared

| Country | Crop | Planting Period | Mid-Season | Harvest Period |
|---|---|---|---|---|
| Bhutan | Corn | Feb–Mar | Apr–Jun | Jul–Sep |
| India | Corn (Kharif) | Mar–Jun | Jul–Aug | Sep–Oct |
| India | Cotton | Apr–Jul | Aug–Sep | Oct–Dec |
| India | Millet (Kharif, Pearl) | May–Jul | Aug | Sep–Nov |
| India | Peanut (Kharif) | May–Jul | Aug | Sep–Nov |
| India | Rice (Kharif) | May–Jul | Aug | Sep–Nov |
| India | Sorghum (Kharif) | May–Jul | Aug | Sep–Oct |
| India | Soybean | Jun–Jul | Aug | Sep–Oct |
| India | Sunflowerseed (Kharif) | Jun–Jul | Aug | Sep–Oct |
| Nepal | Millet | May–Jul | Aug | Sep–Nov |
| Nepal | Rice | May–Jul | Aug–Sep | Oct–Dec |
| Pakistan | Corn | May–Jul | Aug | Sep–Oct |
| Pakistan | Cotton | Mar–Jun | Jul–Aug | Sep–Nov |
| Pakistan | Millet | May–Jun | Jul | Aug–Sep |
| Pakistan | Peanut | Mar–Jun | Jul | Aug–Oct |
| Pakistan | Rice | May–Jul | Aug | Sep–Nov |
| Pakistan | Sorghum | Jun–Jul | Aug | Sep–Oct |
| Pakistan | Sunflowerseed | Jan–Feb | Mar–May | Jun |

**Table 2.** Crop calendars for selected Kharif crops in Bhutan, India, Nepal, and Pakistan from USDA.

to May could be attributed to meteorological factors. The monsoon season in South Asia results in increased wet deposition, and notably, 2019 experienced the most intense monsoon since 1994 (NASA, 2020). As shown in the Fig. S6 (a) and (b), precipitation and temperature in July are the highest of the year. High temperature could increase ammonia volatilization, leading to higher concentrations, while high precipitation increases the wet deposition of ammonia. However, the impact of temperature on concentration is secondary compared to the dramatic variations in precipitation. These combined factors result in July having a smaller concentration peak compared to May, despite July being another peak month. Additionally, CrIS also exhibits minor peaks in May and July, consistent with our posterior results.

Fig. 5 (a-i) illustrates the annual average $NH_3$ column concentrations observed by the IASI satellite instruments from 2015 to 2023. The data clearly show that Pakistan and northern India consistently experience the highest $NH_3$ concentrations, with values exceeding $5 \times 10^{16}$ molec cm$^{-2}$. Furthermore, the spatial distribution of annual average $NH_3$ column concentrations has remained relatively stable over the past decade.

Fig. 5 (j) depicts the monthly mean $NH_3$ column concentrations derived from the IASI satellite. The time series reveals a clear seasonal pattern, with peaks occurring in summer and lower levels in winter, and the highest concentrations consistently observed in July. Additionally, the inter-annual variation in $NH_3$ column concentrations from 2013 to 2019 exhibits a modest

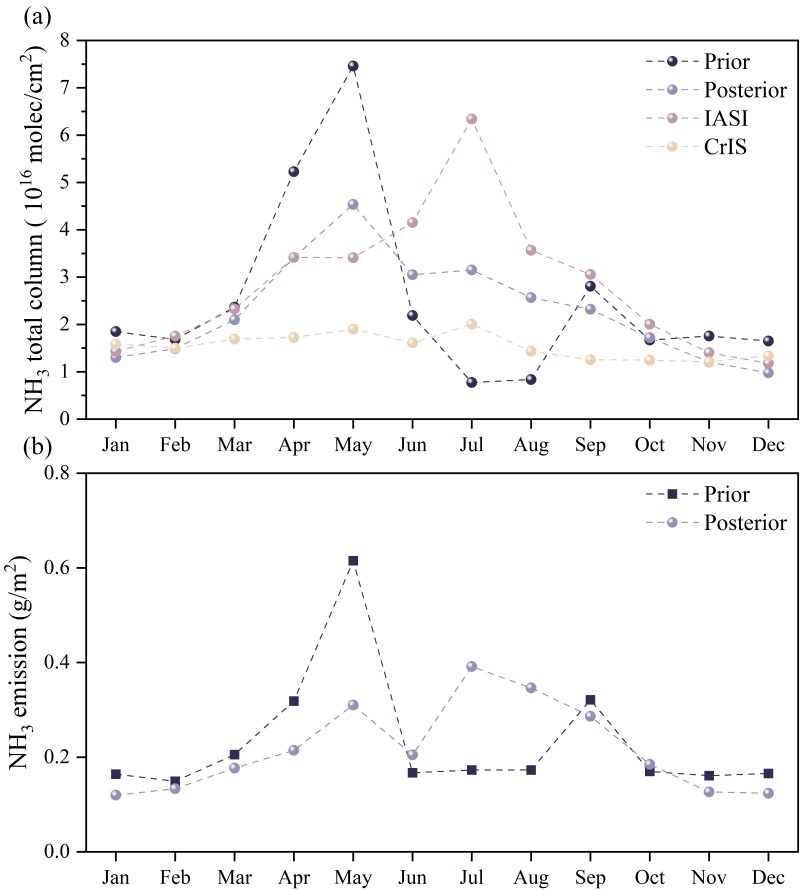

**Figure 4.** The monthly average total NH$_3$ column concentrations from the prior and posterior, IASI-observed, and CrIS-observed from January to December (a). The monthly average values of prior and posterior emissions from January to December (b).

upward trend, ranging from 2.17 to 2.6 ($\times$ 10$^{16}$ molec cm$^{-2}$), corresponding to an average growth rate of approximately 6.32%. Subsequent to 2019, NH$_3$ concentrations stabilize within the range of 2.6 to 2.8 ($\times$ 10$^{16}$ molec cm$^{-2}$). Given the relatively stable NH$_3$ levels after 2019, we restricted our analysis to conducting an assimilation-based emission inversion for the year 2019. Extending emission inversion over a longer period would require substantial computational resources.

## 3.2 Anthropogenic NH$_3$ emissions analysis

By assimilating IASI NH$_3$ column concentrations, the posterior anthropogenic monthly NH$_3$ emission inventories for 2019 were updated. Scenarios of the posterior emission inventories, along with the increments (posterior minus prior), for January, April, July, and November are shown in Fig. 3 (b)-(c). The prior, posterior, and increment data for the remaining months of 2019 are provided in Fig. S7-S9 in the Supplementary Material. Our posterior inventory demonstrated that the primary sources of NH$_3$ originated from North India, Pakistan, and Bengal in general. This finding is consistent with the CEDS

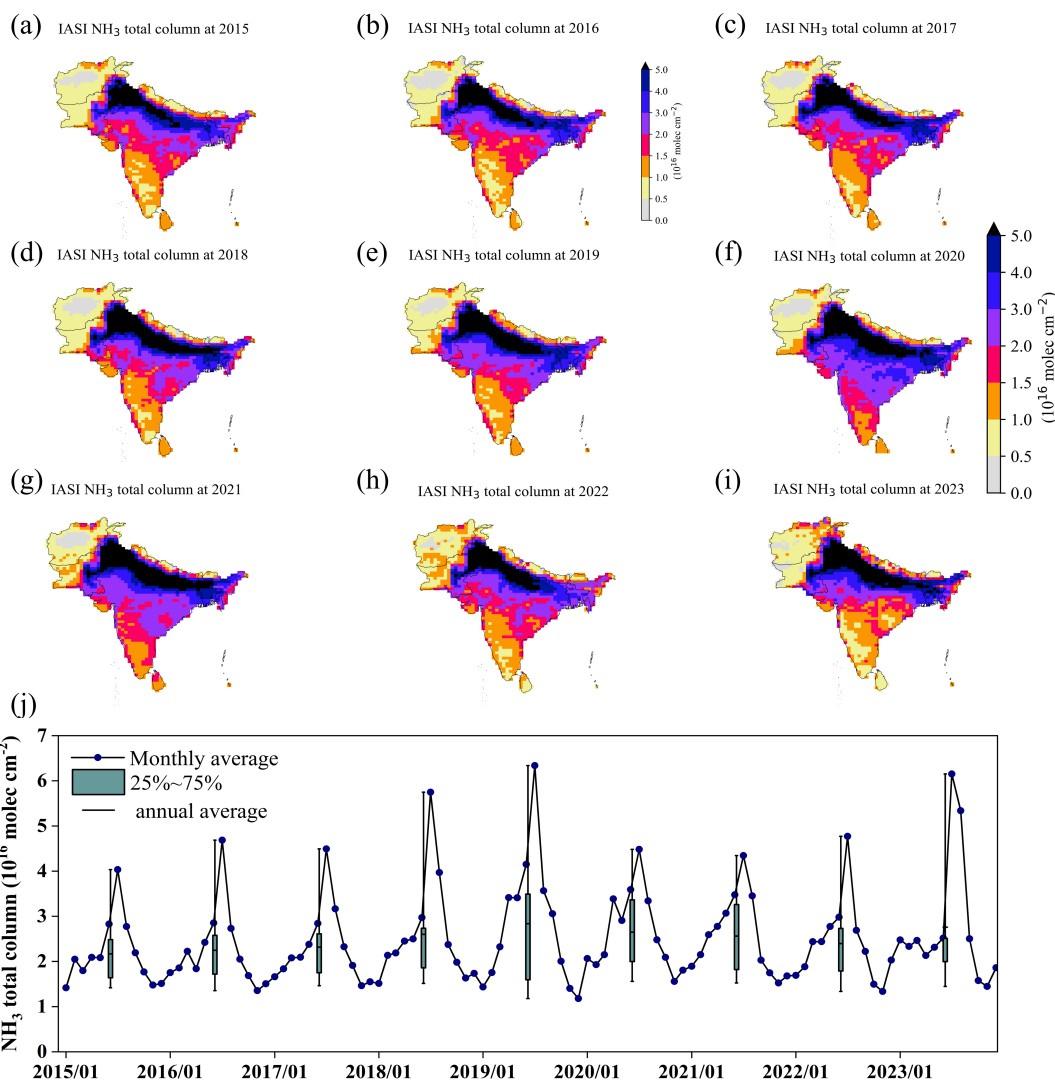

**Figure 5.** The spatial distribution of the annual averaged IASI column concentrations in South Asia from 2015 to 2023 is shown in panels (a) to (i). Panel (j) presents a time series depicting the monthly variation in IASI-observed NH$_3$ column concentrations from 2015 to 2023, with the box plots representing the yearly averages showing interannual changes.

inventory, as well as with other studies (Pawar et al., 2021a). However, huge discrepancy are presented when we compared the posterior and prior result, particular in April in Fig. 3(b) and July in Fig. S3 (c). The posterior results reveal a distinct seasonal emission profile compared to the prior. Specifically, emissions during the spring are significantly overestimated in the prior model, whereas summer emissions are underestimated by up to threefold.

To better illustrate the differences in timing profiles throughout the year, the monthly average emission intensity over South Asia was calculated and is shown in Fig. 4 (b). The prior anthropogenic emission inventory exhibits a "double-peak" pattern, mirroring the profile of the average $NH_3$ concentration displayed in Fig. 4 (a). The emission flux reaches its maximum in May, peaking at approximately 0.6 $g/m^2$, with a secondary peak occurring in September around 0.25 $g/m^2$. In contrast, the assimilation that integrates prior CEDS emissions with IASI measurements shows much lower intensities from January to May,
with the largest negative differences (> 0.3 $g/m^2$) observed in May. While the prior emissions remain relatively low during the summer, the emission inversion reveals positive increments, with the posterior inventory indicating the maximum emission flux in July, peaking at approximately 0.4 $g/m^2$. In general, the posterior emissions also display a "double-peak" pattern; however, the peaks occur in May and July, in contrast to the May and September peaks observed in the prior emissions.

    The substantial emissions in July, as indicated by the posterior anthropogenic inventory, can be attributed to the increased
fertilizer application for crops during the summer season (Tanvir et al., 2019). As shown in Table 2, the sowing period for crops in South Asia is generally from May to July, with July being the peak growth period for crops, resulting in a large amount of fertilization, resulting in July surpassing May in emission intensity. From July to September, as rice and other crops progress through their growth stages, fertilizer application typically decreases, leading to a gradual reduction in $NH_3$ emissions. Additionally, temperatures decline from August to September Fig. S6 (b), reducing the volatilization rate of $NH_3$,
thereby leading to a further decrease in emissions. This pattern occurs because $NH_3$ volatilization is strongly influenced by temperature (Fan et al., 2011).

    The convergence of prior and posterior anthropogenic emission intensities in June is attributed to the overall offsetting of negative and positive increments in the region, as shown in Fig. S9 (f). As depicted in panel (c) of Fig. 3, the negative increments observed in January and April primarily originate from the Indian region, while the positive increments in July and
September are predominantly observed in the same area. Additionally, the posterior emission estimates, which are based on CrIS, have now been included in

### 3.3   Validation

    To evaluate our inversion results, we here compared the atmospheric $NH_3$ simulation either driven by the posterior emission (refer to as the posterior simulation), or driven by the prior one against the observations, including the assimilated IASI
column data, the independent CrIS retrieval and ground-based $NH_3$ and $PM_{2.5}$ concentration measurements.

### 3.3.1   $NH_3$ total column concentration validation

    The difference between the model and IASI observations for the entire year of 2019 is shown in Fig. 6 (a). The overestimation by the prior model is particularly evident in Spring (especially May), while the underestimation is most prominent

in Summer (especially July). These discrepancies contributed to a relatively high model error, with the correlation coefficient (R) as low as 0.33 and the root mean square error (RMSE) as high as $4.64 \times 10^{16}$ molec cm$^{-2}$. In contrast, the posterior emission-driven GEOS-Chem simulations showed good consistency with the IASI retrievals, capturing both the spatial and temporal variations, as shown in panel (d) of Fig. 1. This resulted in significantly improved performance, with R increasing to 0.76 and RMSE reducing to $2.48 \times 10^{16}$ molec cm$^{-2}$, as shown in panel (b) of Fig. 6. The discrepancy between the model and the posterior results mentioned in Section 3.1.1 in July is also evident in the scatter plot of the posterior column simulation against the IASI measurements in panel (b) of Fig. 6.

In addition, we further evaluated our posterior simulations using the other advanced satellite NH$_3$ product from the CrIS instruments. The scatter plots of the CrIS monthly NH$_3$ column concentrations vs. the prior/posterior simulations in 2019 are presented in panels (c) and (d) of Fig. 6. Steady improvements were observed in the comparison against the independent CrIS retrievals, with the correlation coefficient (R) increasing from 0.42 to 0.71, and the root mean square error (RMSE) decreasing from 3.96 to $2.06 \times 10^{16}$ molec cm$^{-2}$. These evaluations give us confidence that our emission inversion has successfully calculated the most likely posterior, given both the prior and the IASI measurements.

### 3.3.2   NH$_3$ and PM$_{2.5}$ ground concentration validation

The few surface NH$_3$ concentration observations from ground stations, shown in Fig. 2, were also utilized to evaluate our NH$_3$ emission inversion results. Fig. 7 presents the scatter plot of monthly surface NH$_3$ concentrations against the prior/posterior simulations. Our posterior results are in better agreement with these independent surface NH$_3$ concentration measurements. This is evident from the higher correlation coefficient (R = 0.39) in the posterior compared to R = 0.28 in the prior simulation. The RMSE values remained almost the same, changing slightly from 22.18 $\mu$g/m$^3$ in the prior to 22.73 $\mu$g/m$^3$ in the posterior. The large remaining error is due to several instances where ground NH$_3$ concentration measurements indicated values several times higher than our simulations. This was also reported by Pawar et al. (2021b), which suggest that ground NH$_3$ observations are likely to overestimate NH$_3$ levels. The mismatch between ground observations and simulations may be attributed to the fact that most monitoring stations are located in urban regions of India, where NH$_3$ concentrations are higher due to traffic and human activities (Sharma et al., 2014). Simulations with an extremely fine resolution could provide a more accurate representation of NH$_3$ characteristics at these surface sites. However, such simulations would significantly increase the computational burden on the emission inversion system, which is beyond the scope of this study.

NH$_3$ is the key precursor of the inorganic aerosol. The estimated NH$_3$ emission inventory is supposed to improve the aerosol simulation as well, under the assumption that aerosols from other sources are accurately represented. The monthly averaged PM$_{2.5}$ concentrations against the simulations either using our prior or using the posterior NH$_3$ inventory, as shown in Fig. 8 (a-b). It is evident that both RMSE and Bias have been reduced to varying degrees: RMSE decreased from 29.15 $\mu$g/m$^3$ in the prior to 22.75 $\mu$g/m$^3$ in the posterior, and bias decreased from 24.8 $\mu$g/m$^3$ in the prior to 18.37 $\mu$g/m$^3$ in the posterior. These results indicate that the emission inventory optimized by our inversion system has improved the model's performance in simulating PM$_{2.5}$, reducing both systematic biases and model underestimation effectively.

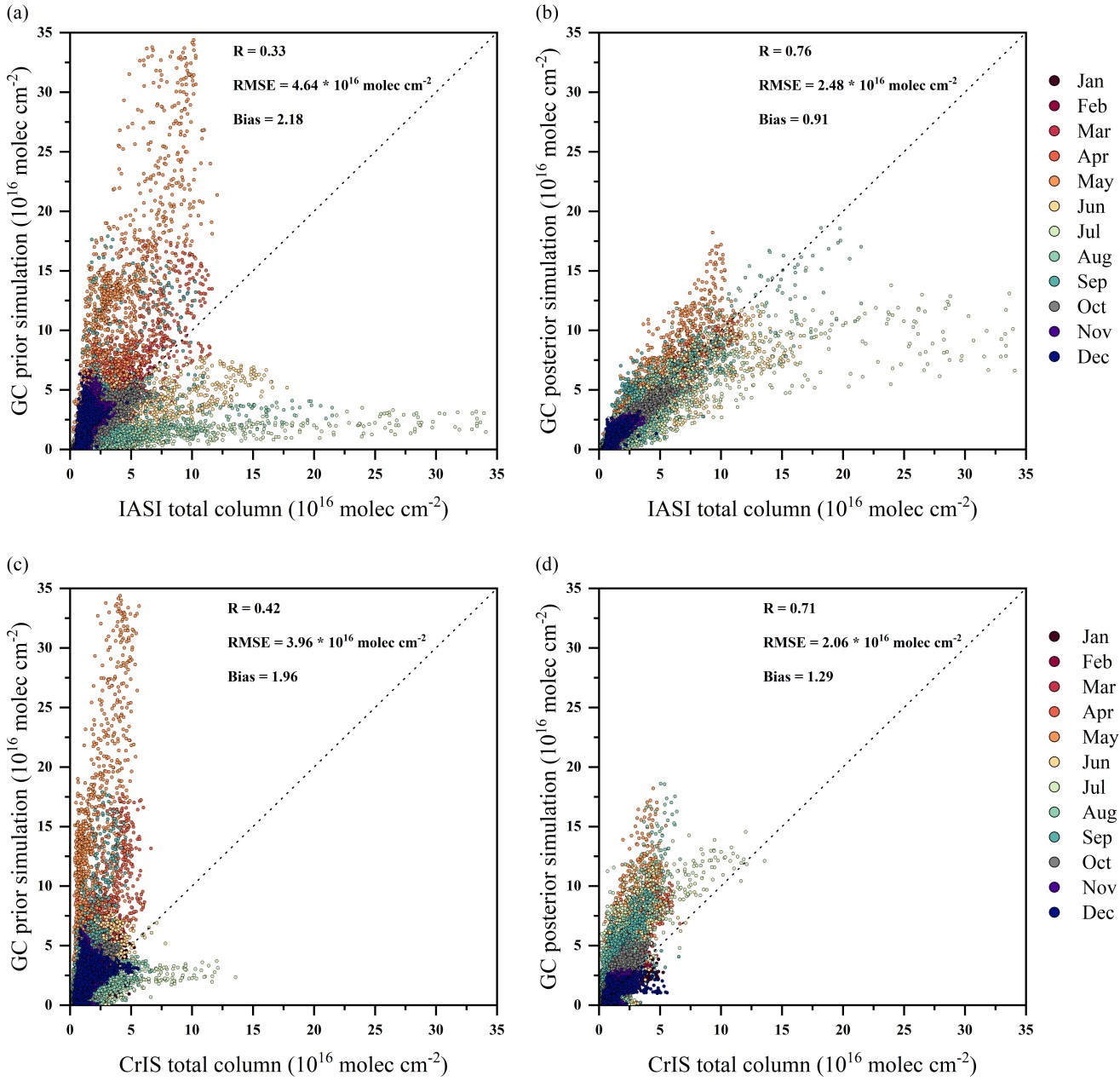

**Figure 6.** Scatter plot of the IASI (a-b) and CrIS (c-d) observed NH$_3$ concentrations against the NH$_3$ simulation over South Asia, either using the prior or the posterior NH$_3$ emission inventory, from January to December.

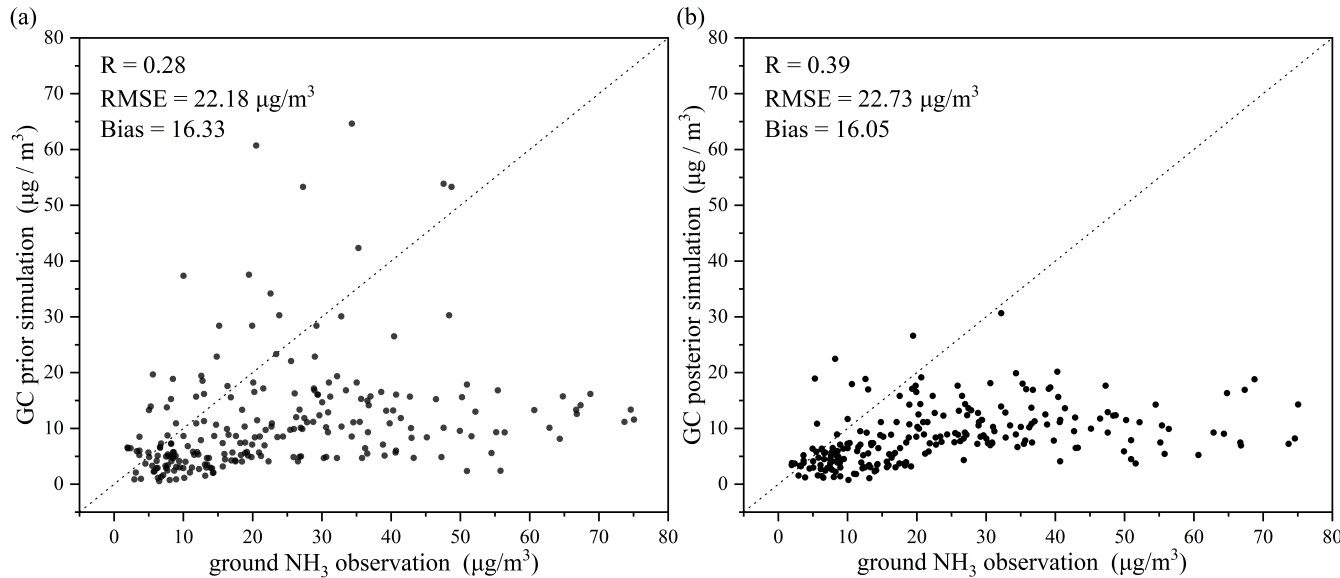

**Figure 7.** Scatter plot of the ground-observed against the NH₃ simulation over South Asia either using the prior (a) or using the posterior (b) NH₃ emission inventory in 2019.

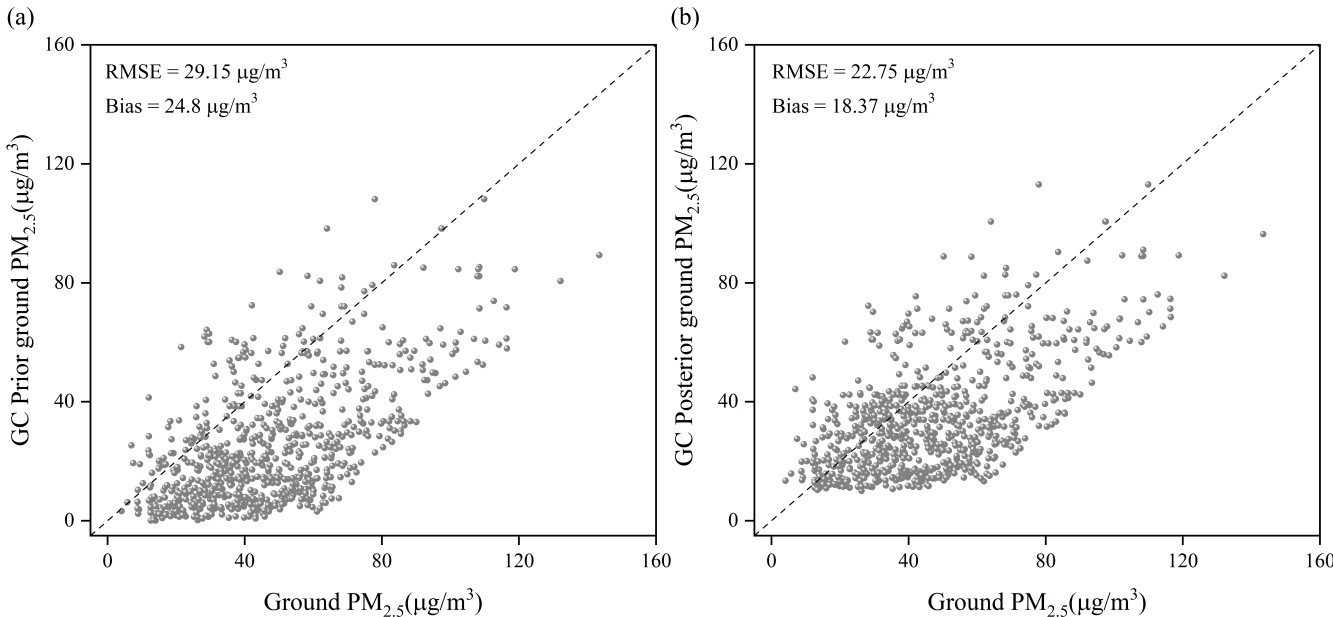

**Figure 8.** Scatter plot of the ground-observed against the PM₂.₅ simulation over South Asia either using the prior (a) or using the posterior (b) emission inventory in 2019.

## 4 Summary and conclusion

South Asia has been severely affected by NH$_3$, which has significant impacts on both human health and the ecological environment. The current emission inventories, primarily based on bottom-up approaches, are subject to substantial uncertainties. This is due to the fact that the intensity of NH$_3$ emissions from livestock and fertilizers is heavily influenced by management and farming practices, yet this information is often not widely available. As a result, accurately simulating the spatiotemporal characteristics of atmospheric NH$_3$ and evaluating its impacts remains challenging. The use of satellite observations, such as those from IASI, for top-down emission inversion has emerged as an effective method to develop more accurate inventories. However, research in this area remains limited in South Asia.

This study employed a 4DEnVar-based emission inversion system to optimize anthropogenic NH$_3$ emissions in South Asia. The most likely posterior monthly anthropogenic NH$_3$ emission inventories were calculated given the the CEDS prior inventory and the NH$_3$ column concentration observations from the polar-orbiting IASI satellite instrument. Validation against satellite and ground-based observations shows that NH$_3$ concentration simulations driven by the posterior emissions perform significantly better than those driven by the prior inventory. In the comparison against the IASI measurments. the correlation coefficient (r) increased from 0.33 (for the prior) to 0.76, and the root mean square error (RMSE) was reduced from 4.64 × 10$^{16}$ molec cm$^{-2}$ (prior) to 2.48 × 10$^{16}$ molec cm$^{-2}$ (posterior). The posterior results also show improvements when compared to independent CrIS satellite measurements, with the correlation coefficient (r) rising from 0.42 (prior) to 0.71, and RMSE reducing from 3.96 × 10$^{16}$ molec cm$^{-2}$ (prior) to 2.06 × 10$^{16}$ molec cm$^{-2}$ (posterior). Additionally, validation with ground-level NH$_3$ and PM$_{2.5}$ concentrations further supports the findings, demonstrating that our emission inversion system effectively reduces systematic biases and underestimation in ground-level simulations.

The spatial and temporal characteristics of anthropogenic NH$_3$ emissions over South Asia were then analyzed based on the inversion. While the prior CEDS inventory generally captured the NH$_3$ emission hotspots, such as in Pakistan, North India, and Bengal, it failed to accurately represent the seasonal trend. Specifically, the prior inventory showed a "double-peak" pattern throughout the year, with peaks in May and September. In contrast, the posterior results revealed the correct seasonal pattern with the "double-peak" profile occurring in May and July. The posterior emission inventory's total annual estimate is 12.61 Tg, compared to the prior inventory's 13.32 Tg.

The top-down NH$_3$ emission inversion system driven by IASI observations has demonstrated superior performance in enhancing the NH$_3$ emission estimates. Nevertheless, several challenges persist, such as the requirement for simulations at finer resolutions to precisely capture very local emission dynamics. Furthermore, observations from stationary satellites, such as FY-4B, also deserve attention for exploring the diurnal variations of the NH$_3$ emission. Our next steps will focus on further refining the spatiotemporal patterns at the daily or weekly scale, building on the current posterior results."

**Code and data availability**

The NH$_3$ emission inversion system is in the Python environment and is archived on Zenodo (https://doi.org/10.5281/zenodo.7015397). The NH$_3$ prior and posterior emission inventories are archived on Zenodo (https://doi.org/10.5281/zenodo.

14979151). The IASI ANNI-NH$_3$-v4R-ERA5 data suites are available at https://iasi.aeris-data.fr/. The CrIS v1.6.4 data are available at https://hpfx.collab.science.gc.ca/~mas001/satellite_ext/cris/. The observed NH$_3$ and PM$_{2.5}$ concentrations data is available at https://www.kaggle.com/datasets/abhisheksjha/time-series-air-quality-data-of-india-2010-2023?select=AP001. csv.

## 5 Acknowledgments

This work is supported by the National Natural Science Foundation of China (Grant No.42475150), Natural Science Foundation of Jiangsu Province (Grant No. BK20220031), The National Natural Science Foundation of China (Grant No.42305194) and Natural Science Foundation of the Higher Education Institutions of Jiangsu Province (22KJB630012).

## Author contribution

JJ designed the study. JX performed the data analysis, produced the figures, and drafted the initial manuscript. YZ contributed to the model simulations. All the authors contributed to the discussion and editing of the manuscript.

## Competing interests

The authors declare that they have no conflict of interest.

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
