# Peer review of "South Asia ammonia emission inversion through assimilating IASI observations"

_EGUsphere, 2024_

## Author Comment (AC1)

**Authors' Response to Reviews of**

**South Asia ammonia emission inversion through assimilating IASI observations**

Ji Xia[*], Yi Zhou[*], Li Fang, Yingfei Qi, Dehao Li, Hong Liao, and Jianbing Jin
*Atmospheric Chemistry and Physics Discussions,*
* * *
RC: Reviewers' Comment, AR: Authors' Response, □ Manuscript Text

**General Comments:**

RC: *This paper focuses on optimizing the CEDS* NH₃ *emission inventory using a top-down inversion system (4DEnVar) over South Asia. The authors have previously conducted a similar work in China (published in ERL), but they highlight several novelties in this study:*

1. *The application of the IASI averaging kernel to derive* NH₃ *column concentrations;*
2. *A relatively high spatial resolution* NH₃ *emission inversion over South Asia (0.5° × 0.625°);*
3. *The identification of a "double-peak" seasonal pattern difference between prior and posterior emissions.*

*Overall, while the study's scope aligns well with the journal ACP and contributes to advancing the field by improving top-down* NH₃ *emissions over South Asia*

Response to Referee #1: We would like to thank the referee for the careful review throughout the paper and the in-depth comments that help to improve our paper.

**Specific Comments:**

RC: *1) Clarify the use of observations and simulations: The use of IASI and CrIS observations, along with GEOS-Chem simulations, should be more clearly outlined. For example, a table could be included summarizing the exact periods covered by each dataset (i.e., 2015-2023, 4 months in 2019), their respective roles in the study, and the purposes they serve. Additionally, the data filtering process for removing irrational values should be more rigorously justified, particularly regarding skin temperature and cloud fraction considerations. Comparisons between IASI, CrIS, or previous studies could help validate the dataset selection.*

AR: Thanks for comment. We have integrated the relevant information into a table and added it to the manuscript. Additionally, we have added the previously omitted cloud fraction filtering

information in the manuscript, and re-calculated the IASI dataset using the skin temperature condition. Details are below:

*Text in manuscript*

The assimilated observations for estimating the $NH_3$ emissions were the monthly IASI column concentration means over the 0.5 °× 0.625 °GEOS-Chem grid cell. These values were derived from the latest ANNI-$NH_3$-v4R-ERA5 product. Despite improvements in $NH_3$ column retrievals from satellite observations, there remains substantial variability in measurement uncertainty, ranging from 5 % to over 1000 %. (Van Damme et al., 2014b; Whitburn et al., 2016; Van Damme et al., 2017). Data selection was performed by excluding nighttime observations, irrational values (<0), and only using data with a cloud fraction < 0.1 (Van Damme et al., 2018) and skin temperature > 263 K (Van Damme et al., 2014a) during the calculation of the monthly mean. It is important to note that the time coverage of the available version 4 IASI product used was limited: Metop-A provided data for the entire year of 2019, Metop-B provided data from January to July 2019, and Metop-C did not have data for 2019. Therefore, only the data from Metop-A and Metop-B within the 2019 time frame were used in this study.

| Data | Period | Use |
|---|---|---|
| IASI v3 | 2015-2023 | Annual variation of $NH_3$ concentration |
| IASI v4 | entire 2019 | Inversion and Validation |
| Level 2 CrIS | entire 2019 | Independent validation |
| CPCB | entire 2019 | Independent validation |
| GEOS-Chem | entire 2019 | Similation |

**Table 1.** The use of observations and simulations

**RC:** *2) Present results in a logical sequence: The results section could benefit from a clearer structure. It might be more intuitive to first present the analysis of observed $NH_3$ concentrations (Sec. 3.1.1 and 3.2) before discussing the spatial and temporal patterns of $NH_3$ emissions. Then followed by the validation with the concentration inferred using the inversion.*

**AR:** Thanks for the in-depth comment. In response to your suggestion, I have revised the logical sequence of the sections in the manuscript. The following is the revised section sequence:

*Text in manuscript*

3 Results and discussion
3.1 Observed $NH_3$ concentrations
3.1.1 Spatial $NH_3$ total column concentration

**RC:** *3) Strengthen the discussion: The discussion should delve deeper into the reasons behind the changes in the seasonal patterns observed in the posterior emissions compared to the prior. Supporting this analysis with additional figures in either the main text or supplementary materials would be valuable. For instance, including information on:*

- *Seasonal fertilizer application timings for major crops like rice, corn, and wheat;*
- *Meteorological factors affecting $NH_3$ volatilization and deposition, such as temperature and precipitation;*
- *Biomass burning patterns in 2019, as they may influence spatial and seasonal variations of $NH_3$ emissions and concentrations.*

**AR:** Appreciate your comment. We have added the meteorological conditions, such as precipitation and temperature, as well as the seasonal variations in biomass burning, to the supplementary materials. Additionally, the trend of anthropogenic ammonia emissions is mainly attributed to agricultural activities, so the changes in anthropology emissions generally correspond to the planting times of crops. We have identified the planting and harvesting times of crops in the South Asia region from USDA(U.S.DEPARTMENT OF ARGRICULTURE, https://ipad.fas.usda.gov/rssiws/al/crop_calendar/sasia.aspx). The detailed emission analysis in revised manuscript is as follows:
* * *
*Text in manuscript*

3.1.2 *Seasonal and annual variation of $NH_3$ concentration*

…

The high value in May is attributed to huge amount of biomass burning in South Asia during the spring in Figure S4 (c). We have identified the planting and harvesting times of crops in the South Asia region from USDA(U.S.DEPARTMENT OF ARGRICULTURE, https://ipad.fas.usda.gov/rssiws/al/crop_calendar/sasia.aspx). The heavy use of fertilizers in agricultural activities has resulted in the highest emission throughout the year, as will be illustrated in Fig. 4 (b) in Section 3.2. This has lead to the second $NH_3$ concentration peak in July. The reasons for higher emissions in July but lower concentration levels compared to May could be attributed to meteorological factors. The monsoon season in South Asia results in increased wet deposition, and

notably, 2019 experienced the most intense monsoon since 1994 (NASA, 2020). As shown in the Figure S4 (a) and (b), precipitation and temperature in July are the highest of the year. High temperatures increase ammonia volatilization, and the high precipitation increases the wet deposition of ammonia. These combined factors lead to July having a smaller concentration peak compared to May, despite being another peak month.

…

**3.2 Spatial and Seasonal variation of NH₃ emission**

…

The substantial emissions in July, as indicated by the posterior inventory, can be attributed to the increased fertilizer application for rice and corn crops during the summer season (Tanvir et al., 2019). Although biomass burning emissions are generally higher in spring in Figure S4 (c), agricultural activities remain the primary contributors to $NH_3$ emissions (Huang et al., 2016), resulting in July surpassing May in emission intensity. From July to September, as rice and other crops progress through their growth stages, fertilizer application typically decreases, leading to a gradual reduction in $NH_3$ emissions. Additionally, temperatures decline from August to September in Figure S4 (b), reducing the volatilization rate of $NH_3$. This pattern occurs because $NH_3$ volatilization is strongly influenced by temperature (Fan et al., 2011)

[Figure]

Figure 4. The monthly average total $NH_3$ column concentrations from the prior and posterior, IASI-observed, and CrIS-observed from January to December (a). The monthly average values of

prior and posterior emissions from January to December (b).

*Supplement*

[Figure]

Figure S4. Monthly precipitation (a), temperature (b) from MERRA2 and biomass buring
emission (c) from GFED4 in 2019.

**RC:** *4) Abstract: One interesting point of this study is the different 'double-peak' seasonal pattern from the CEDS prior. I would expect to highlight this point in the abstract and briefly talk about the potential reasons for this.*

**AR**: Thanks for the comment. I have revised the abstract to highlight the interesting finding of the "double-peak" seasonal pattern observed in this study, which differs from the CEDS prior. Additionally, I briefly discuss the potential reasons for this discrepancy in the revised abstract. The following is the revised abstract excerpt:

*Text in Abstract:*

Notably, emissions there exhibit a "double-peak" seasonal profile, with the maximum in July and the secondary peak in May. This differs from the "double-peak" trend suggested by the CEDS prior inventory, which identifies the maximum in May and a second peak in September. The differences may be attributed to a more accurate representation of regional agricultural practices, such as the

timing of fertilizer application and meteorological influences like precipitation and temperature.

**RC:** *5) The effect of NH₃ on climate change: inversed cause and effect, Sanderson et al., 2006 studied the effect of climate change on acid deposition. Consider reading more related and updated papers:*

*Gong, Cheng, et al. "Global net climate effects of anthropogenic reactive nitrogen." Nature 632.8025 (2024): 557-563.*

*Ma, Ruoya, et al. "Data-driven estimates of fertilizer-induced soil NH₃, NO and N2O emissions from croplands in China and their climate change impacts." Global Change Biology 28.3 (2022): 1008-1022.*

**AR**: Many thanks for your feedback. I have reviewed the recommended papers and other related recent studies, and I have updated the manuscript accordingly blew:

*Text in manuscript:*

1 *Introduction*

Ammonia ($NH_3$), an alkaline compound, has the capacity to react with acidic gases present in the atmosphere, thereby contributing to the formation of secondary aerosols, notably ammonium sulfate and ammonium nitrate (Jimenez et al., 2009). The genesis of fine atmospheric particulate matter poses significant threats to human health (Mukherjee and Agrawal, 2017). Further, ammonia gas, along with its reaction products, plays a pivotal role in soil acidification and the eutrophication of water bodies through both dry and wet deposition (Krupa, 2003), and thereby affecting the balance of ecosystems (Asman et al., 1998) and climate change (Ma et al., 2022; Gong et al., 2024).

**RC:** *6) NH₃ level over South Asia, I would expect to see the number of* $NH_3$ *concentrations or emissions here, as stated the highest in the world.*

**Reply**: I have updated the manuscript to include the specific $NH_3$ concentration data for South Asia. The specific revised excerpt is as follows.

*Text in manuscript:*

1 *Introduction*

…

With an enormous livestock population and extensive use of nitrogen fertilizers, South Asia has

experienced the highest level of atmospheric $NH_3$ globally (Pawar et al., 2021b; Luo et al., 2022). Specifically, the annual average ammonia concentration across India is approximately $1.8–5.6 \times 10^{16}$ mol/cm$^2$, while in the Indo-Gangetic Plain (IGP) of India, the concentration is double that of other regions, reaching a peak of $11.5 \times 10^{16}$ mol/cm$^2$ during the high season in July (Kuttippurath et al., 2020).

**RC: 7) Use NH₃ instead of ammonia after the definition, which applies to the whole text.**

**AR**: Appreciate your comment. I have made the necessary revisions and replaced "ammonia" with "$NH_3$" throughout the manuscript, following the initial definition.

**RC: 8) "the environmental impacts can be quantified" -> "enabling the quantification of environmental impacts"**

**AR**: Thanks for your comment. I have revised the sentence as per your recommendation, changing "the environmental impacts can be quantified" to "enabling the quantification of environmental impacts." The specific revised excerpt is as follows:

**Text in manuscript:**

1 **Introduction**

…

Over the past decade, scientists have predominantly employed the "bottom-up" approach to estimate $NH_3$ emissions. When combined with chemical transport models, atmospheric $NH_3$ dynamics can be simulated, enabling the quantification of environmental impacts.

….

**RC: 9) however, these bottom-up estimates of NH₃ emissions are generally considered uncertain (Xu et al., 2019), especially compared to other pollutants mainly derived from fossil fuel combustion" -> However, these bottom-up estimates of NH₃ emissions are generally considered as uncertain (Xu et al., 2019), particularly when compared to pollutants primarily originating from fossil fuel combustion, such as ...**

**AR**: Thanks for comment. I have revised the sentence as recommended, changing it to: "However, these bottom-up estimates of $NH_3$ emissions are generally considered as uncertain (Xu et al., 2019), particularly when compared to pollutants primarily originating from fossil fuel combustion, such

as...". The specific revised excerpt is as follows:

*Text in manuscript:*
* * *
1 *Introduction*

…

Over the past decade, scientists have predominantly employed the "bottom-up" approach to estimate $NH_3$ emissions. When combined with chemical transport models, atmospheric $NH_3$ dynamics can be simulated, enabling the quantification of environmental impacts. Substantial efforts have been made to quantify the spatiotemporal distribution of $NH_3$ sources and develop global/regional emission inventories, such as the global $NH_3$ emission inventory (Bouwman et al., 1997), the anthropogenic emission inventory that includes $NH_3$ estimates (e.g., Community Emissions Data System, CEDS) (Hoesly et al., 2018), as well as regional $NH_3$ inventories focusing on South Asia (Yan et al., 2003; Yamaji et al., 2004; Liu et al., 2022). However, these bottom-up estimates of $NH_3$ emissions are generally considered as uncertain (Xu et al., 2019), particularly when compared to pollutants primarily originating from fossil fuel combustion such as NO2.
* * *
*RC: 10) "limitations still remain": unclear what kind of limitations here.*

**AR**: I have clarified the statement regarding the limitations in the manuscript. The next sentence has been revised to explain more specifically that "These primarily arise from the fact that satellite observations can only measure column-integrated $NH_3$ concentrations, which do not directly reflect emission intensity or the three-dimensional concentration field." The specific revised excerpt is as follows:

*Text in manuscript:*
* * *
1 *Introduction*

…

The rapid advancement of satellite remote sensing technology has resulted in an expanding array of valuable $NH_3$ products, such as those from the first satellite $NH_3$ observations using the Tropospheric Emission Spectrometer (TES) (Beer et al., 2008), as well as higher-resolution retrievals from the Infrared Atmospheric Sounding Interferometer (IASI) (Pawar et al., 2021b) and the Cross-track Infrared Sounder (CrIS) (Beale et al., 2022; Kharol et al., 2022). While these remote sensing measurements play a pivotal role in characterizing atmospheric $NH_3$ loading, limitations still remain. These primarily arise from the fact that satellite observations can only measure column-integrated $NH_3$ concentrations, which do not directly reflect emission intensity or the three-dimensional concentration field. In addition to these satellite-based data, very limited ground-based

observations are publicly available over South Asia, and those that do exist are constrained by their inadequate representation of atmospheric $NH_3$ features (Pawar et al., 2021b).

*RC: 11) Add a citation about your emission inversion system.*

**AR**: Thanks for comment. We have added the relevant citation for our emission inversion system in the manuscript (Jin et al., 2023). The updated sentence now reads: "The $NH_3$ emission inventory will be calculated using our newly developed emission inversion system (Jin et al., 2023)..."

*Text in manuscript:*

1 *Introduction*

…

However, there is a paucity of studies focusing on assimilation-based $NH_3$ emission inversion specific to South Asia, which has some of the highest $NH_3$ loading hotspots compared to other continents. In this study, we aim to explore the spatial and temporal features of $NH_3$ emissions over South Asia. The $NH_3$ emission inventory will be calculated using our newly developed emission inversion system (Jin et al., 2023), by assimilating $NH_3$ retrievals from the IASI instruments onboard 15 MetOp-A (operational from 2008 to 2018), MetOp-B (operational since 2012), and MetOp-C (operational since 2018) satellites.

*RC: 12) The description of emission inversion system and corresponding data could be shortened here and add more details in the data and method part.*

**AR**: Thank you for your valuable comment. The description of the emission inversion system and the corresponding data has been shortened here, and more details have been added in the data and method section. The specific revised excerpt is as follows:

*Text in manuscript:*

1 *Introduction*

…

However, there is a paucity of studies focusing on assimilation-based $NH_3$ emission inversion specific to South Asia, which has some of the highest $NH_3$ loading hotspots compared to other continents. In this study, we aim to explore the spatial and temporal features of $NH_3$ emissions over South Asia. The $NH_3$ emission inventory will be calculated using our newly developed emission inversion system (Jin et al., 2023), by assimilating $NH_3$ retrievals from the IASI instruments

onboard MetOp-A (operational from 2008 to 2018), MetOp-B (operational since 2012), and MetOp-C (operational since 2018) satellites. Instead of directly assimilating IASI measurements as previous studies have done, we incorporated the averaging kernel information from the latest version of the IASI product. This approach allowed us to update the column concentration observations before assimilation. By doing so, we ensure a fairer comparison between the simulated and observed columnar $NH_3$ concentrations, a point that has been emphasized in several studies (Eskes and Boersma, 2003; von Clarmann and Glatthor, 2019), but never implemented in the IASI-based emission inversion.

**RC: 13) Clearly state the aims of this study, focusing on but not only the $NH_3$ emission feature.**

**AR**: Many thanks for your feedback. We have revised the relevant section of the manuscript to clearly state the aims of this study. The specific revised excerpt is as follows:

**Text in manuscript:**

1 **Introduction**

**…**

We aims to provide a more accurate estimation of $NH_3$ emission inventories and to explore their spatial and temporal characteristics across South Asia. Additionally, it serves as a model for effectively calculating atmospheric pollution emissions in regions that have been less studied in the past. The study focuses on $NH_3$ emissions but also contributes to a broader understanding of atmospheric pollution in under-researched regions.

**RC: 14) Which periods are used for MetOp A/B/C, respectively?**

**AR**: The latest v4 version of the IASI data, which includes averaging kernel information, has limited time coverage. Specifically, Metop-A covers the entire year of 2019, Metop-B covers from January to July 2019, and Metop-C has no data for 2019. As a result, we only used the available data from Metop-A and Metop-B within the 2019 time frame. I have made the corresponding changes in the manuscript to reflect this belw:

**Text in manuscript:**

**2.1 IASI satellite measurements**

**…**

The assimilated observations for estimating the $NH_3$ emissions were the monthly IASI column

concentration means over the 0.5 °× 0.625 ° GEOS-Chem grid cell. These values were derived from the latest ANNI-NH$_3$-v4R-ERA5 product. Despite improvements in NH$_3$ column retrievals from satellite observations, there remains substantial variability in measurement uncertainty, ranging from 5 % to over 1000 %. (Van Damme et al., 2014; Whitburn et al., 2016; Van Damme et al., 2017). Data selection was performed by excluding nighttime observations, irrational values (< 0.1 (Van Damme et al., 2018) and skin temperature > 263 K (Van Damme et al., 2014) during the calculation of the monthly mean. It is important to note that the time coverage of the available version 4 IASI product used was limited: Metop-A provided data for the entire year of 2019, Metop-B provided data from January to July 2019, and Metop-C did not have data for 2019. Therefore, only the data from Metop-A and Metop-B within the 2019 time frame were used in this study.

**RC:15) If you are using the L2 data, how do you aggregate it into the GEOS-Chem grid?**

**AR**: The data is processed by reading daily NH$_3$ values and adding them to the corresponding GEOS-Chem grid cells. For each valid data point, if its latitude and longitude fall within the specified range, its NH$_3$ value is added to the appropriate grid cell.

**RC: 16) Except for the irrational values, have you detected any large outliers?**

**AR**: Thanks for your comment. We have not detected any significant outliers, aside from the irrational values that were excluded during the data selection process.

**RC: 17) It is unclear how the B, A_z^a, and m_z coming from in the Eq. 1.**

**AR**: Thanks for your comment. To clarify, in Eq. 1:

$$m_z = \frac{M_z^m - B_z}{M^m - B}$$

$M_z^m$ represents the modeled concentration of NH$_3$ at altitude **z**.
$B_z$ is the background concentration of NH$_3$ at the same altitude.
$M^m$ represents the total modeled concentration of NH$_3$ in the atmosphere.
**B** is the total background concentration.

$$A_z^a = \frac{1}{N} \frac{\hat{X}^a - B}{\hat{X}^{|z} - B}$$

$\hat{X}^{|z}$ represents the a priori (or assumed) concentration of NH$_3$ at altitude **z**.
$B_z$ is again the background concentration at that altitude.

$\hat{X}^a$ is the total a priori concentration.

**N** is a normalization factor, ensuring the matrix $A_z^a$ sums correctly to account for all altitudes. The specific revised description of the relevant part is as follows:

*Text in manuscript:*
* * *
**2.1 *IASI satellite measurements***

**…**

Here, $\hat{X}^m$ represents the IASI column concentration retrieved with model profile. $\hat{X}^a$ denotes the initial IASI column concentration, with the background concentration B. The Aa z values are AVK for each vertical layer, with the model profile $m_z$. More detailed information and the corresponding equations are provided in the supplementary materials equation S8 and S9.

**Supplement**

**…**

$$m_z = \frac{M_z^m - B_z}{M^m - B}$$

here $M_z^m$ represents the modeled concentration of $NH_3$ at altitude z. $B_z$ is the background concentration of $NH_3$ at the same altitude. $M^m$ represents the total modeled concentration of $NH_3$ in the atmosphere. B is the total background concentration.

$$A_z^a = \frac{1}{N} \frac{\hat{X}^a - B}{\hat{X}^{|z} - B_z}$$

here $\hat{X}^{|z}$ represents the a priori (or assumed) concentration of $NH_3$ at altitude z. $B_z$ is again the background concentration at that altitude. $\hat{X}^a$ is the total a priori concentration. N is a normalization factor, ensuring the matrix $A_z^a$ sums correctly to account for all altitudes.
* * *
***RC: 18) I do not see the 'a' in Eq.3, where has it been used?***

**AR**: Appreciate your comment. **α** is a spatially varying tuning factor. It is used to compensate for the uncertainty in ammonia emissions and to adjust the spatial variability in the emission estimates. **α** influences the calculation of the background error covariance matrix **B** through its spatial variation, helping to optimize the ammonia emission estimates.

***RC: 19) How is the parameter 'σ_integrated' calculated in Eq. 6?***

**AR**: Thanks for comment. σ_integrated is the same as σ_total in Equation 2. I have modified and unified the parameter names in both equations.

*Text in manuscript:*
* * *
*2.1 IASI satellite measurements*

…

$$\sigma^{\text{integrated}} = \left\{ \left( \sigma^{\text{instrument}} \right)^2 + \left( \sigma^{\text{representing}} \right)^2 \right\}^{0.5} \tag{2}$$

*2.3 Emission inversion system*

…

$$\mathbf{O}_{i,i} = \min \left( 1.0 \times 10^{16} \text{ molec cm}^{-2}, \sigma^{\text{integrated}} \right)^2 \tag{6}$$
* * *
*RC: 20) Add more information such as the boundary layer condition, the spin-up process, model version...*

**AR**: Many thanks for your feedback. I have added more detailed information, including the boundary layer condition, the spin-up process, and the model version. The manuscript fragment with more information about the model is as follows:

*Text in manuscript:*
* * *
GEOS-Chem, a three-dimensional (3-D) global tropospheric chemistry model, is driven by assimilated meteorological data obtained from the Goddard Earth Observing System (GEOS) at the NASA Data Assimilation Office (DAO) (Bey et al.,2001). GEOS-Chem incorporates a fully integrated chemistry system involving aerosol, ozone, NOx, and hydrocarbons, as described by Park et al. (2004).The wet deposition scheme is explained by Liu et al. (2001) for water-soluble aerosols and by Amos et al. (2012) for gaseous components. Dry deposition is modeled using the resistance-in-series scheme proposed by Wesely and Lesht (1989), as applied by Wang and Jacob (1998). Size-specific aerosol dry deposition follows the approach outlined by Emerson et al. (2020). A nested grid simulation within the GEOS-Chem model v13.4.1 is conducted to simulate the atmospheric environment over South Asia. The nested domain (60°–98°E, 4°–40°N), shown in Fig. 2, has a horizontal resolution of 0.5°latitude by 0.625° longitude, accompanied by 47 vertical layers. The model is driven by meteorological fields from the Modern-Era Retrospective analysis for Research and Applications, Version 2 (MERRA-2) reanalysis dataset provided by the Global Modeling and Assimilation Office (GMAO) at NASA. The model employs a three-months spin-up period to minimize the influence of initial conditions. Lateral boundary conditions for the nested domain are updated every 3 hours using output from the global GEOS-Chem simulation at 2°× 2.5° resolution. Chemical initial conditions are also obtained from the global simulation to ensure consistency.
* * *
*RC: 21) Sect. 2.3.1 and 2.3.2 can be combined.*

**AR**: Thank you for your kind comment. I have merged Sections 2.3.1 and 2.3.2 into a single section:
***GEOS-Chem model and emission inventory***

**RC: 22) Do you run the model for the whole year 2019?**

**AR**: Yes, we ran the model for the entire year of 2019.

**RC: 23) The step that coarse-grained CEDS into 0.5 * 0.625 degs included in the GEOS-Chem or seperately?**

**AR**: Thanks for comment. The step of coarse-graining the CEDS data into 0.5° × 0.625° is done separately and is not included in the GEOS-Chem model. We directly made modifications to the original emission inventory.

**RC: 24) What is the spatial resolution of the CEDS and is there any regional emission inventory with higher resolution over South Asia that you can use? Does CEDS contain both the anthropogenic and natural sectors?**

**AR**: The spatial resolution of the CEDS emission inventory is 0.5° × 0.5°, and it only covers anthropogenic emission data. For the South Asia region, there are higher resolution emission inventories available. For example, MIXv2 is a long-term emission inventory for Asia with a spatial resolution of 0.1°, covering emission data for Southeast and South Asia.

It is important to note, however, that while higher spatial resolution in emission inventories may seem advantageous, it does not necessarily result in better inversion outcomes, as there is no direct correlation between higher resolution and improved inversion accuracy.

**RC: 25) Figure 1: Increase the size of the dots and change the color regime. I suggest using different point styles for different measurements.**

**AR**: Appreciate your comment. I have increased the size of the dots and adjusted the color regime as requested. Additionally, I have used different point styles to differentiate the various measurements in Figure 1. Considering the sequence of figures mentioned in the original text, we have also changed Figure 1 to Figure 2. The specific revised content is as follows:

***Text in manuscript:***

[Figure]

Figure 2. The GEOS-Chem model simulation domain, with dots indicating the locations of ground observation stations from the Central Pollution Control Board (CPCB), India. The three different colored dots represent stations with only PM$_{2.5}$ observations, stations with both PM$_{2.5}$ and NH$_3$ observations, and stations with only NH$_3$ observations, respectively.

*RC: 26) Figure 1: Can you list these stations in the supplementary?*

**AR**: Appreciate your comment. I have added the list of stations in the supplementary material as requested.

*Table in supplement:*

| Station | lon | lat | StationName | City | State |
|---|---|---|---|---|---|
| AP001 | 80.5181667 | 16.5150833 | Secretariat, Amaravati - APPCB | Amaravati | Andhra Pradesh |
| AP005 | 83.3 | 17.72 | GVM Corporation, Visakhapatnam - APPCB | Visakhapatnam | Andhra Pradesh |
| DL001 | 77.152491 | 28.815691 | Alipur, Delhi - DPCC | Delhi | Delhi |
| DL002 | 77.749675 | 28.725645 | Anand Vihar, Delhi - DPCC | Delhi | Delhi |
| DL019 | 77.2005 | 28.6341 | Mandir Marg, Delhi - DPCC | Delhi | Delhi |
| DL020 | 77.030469 | 28.68241 | Mundka, Delhi - DPCC | Delhi | Delhi |
| DL024 | 80.3229863 | 26.4703136 | Nehru Nagar, Delhi - DPCC | Delhi | Delhi |
| HR012 | 77.0667 | 28.4227 | Sector-51, Gurugram - HSPCB | Gurugram | Haryana |
| KA011 | 77.622813 | 12.917348 | Silk Board, Bengaluru - KSPCB | Bengaluru | Karnataka |
| MH008 | 72.82 | 18.91 | Gole Bazar, Katni - MPPCB | Katni | Madhya Pradesh |
| PB001 | 74.876512 | 31.62 | Golden Temple, Amritsar - PPCB | Amritsar | Punjab |
| RJ004 | 75.836858 | 26.902909 | Adarsh Nagar, Jaipur - RSPCB | Jaipur | Rajasthan |
| RJ005 | 75.7994901 | 26.9164092 | Police Commissionerate, Jaipur - RSPCB | Jaipur | Rajasthan |
| RJ006 | 75.730943 | 26.9502929 | Shastri Nagar, Jaipur - RSPCB | Jaipur | Rajasthan |
| TN001 | 80.1076538 | 12.9099161 | Alandur Bus Depot, Chennai - CPCB | Chennai | Tamil Nadu |
| TG006 | 78.451437 | 17.349694 | Zoo Park, Hyderabad - TSPCB | Hyderabad | Telangana |
| UP012 | 80.9302753 | 26.8821003 | Central School, Lucknow - CPCB | Lucknow | Uttar Pradesh |
| UP013 | 81.005119 | 26.86812 | Gomti Nagar, Lucknow - UPPCB | Lucknow | Uttar Pradesh |
| WB007 | 88.3638022 | 22.5367507 | Ballygunge, Kolkata - WBPCB | Kolkata | West Bengal |
| WB008 | 88.41002457 | 22.58157048 | Bidhannagar, Kolkata - WBPCB | Kolkata | West Bengal |
| AS001 | 91.78063 | 26.181742 | Railway Colony, Guwahati - APCB | Guwahati | Assam |
| MH012 | 72.8204 | 19.3832 | Vasai West, Mumbai - MPCB | Mumbai | Maharashtra |
| WB011 | 88.380669 | 22.627847 | Rabindra Bharati University, Kolkata - WBPCB | Kolkata | West Bengal |
| ML001 | 91.8985 | 25.5586 | Lumpyngngad, Shillong - Meghalaya PCB | Shillong | Meghalaya |
| BR005 | 85.043586 | 25.586562 | DRM Office Danapur, Patna - BSPCB | Patna | Bihar |
| BR006 | 85.227158 | 25.592539 | Govt. High School Shikarpur, Patna - BSPCB | Patna | Bihar |

**Table S1.** Stations with both PM$_{2.5}$ data and NH$_3$ data

| Station | lon | lat | StationName | City | State |
|---|---|---|---|---|---|
| AP002 | 81.7363176 | 16.9872867 | Anand Kala Kshetram, Rajamahendravaram - APPCB | Rajamahendravaram | Andhra Pradesh |
| BR001 | 84.9994 | 24.7955 | Collectorate, Gaya - BSPCB | Gaya | Bihar |
| BR003 | 85.2459 | 25.697189 | Industrial Area, Hajipur - BSPCB | Hajipur | Bihar |
| HR001 | 76.778328 | 30.379589 | Patti Mehar, Ambala - HSPCB | Ambala | Haryana |
| HR002 | 76.9254 | 28.6701 | Arya Nagar, Bahadurgarh - HSPCB | Bahadurgarh | Haryana |
| HR003 | 77.319699 | 28.3419248 | Nathu Colony, Ballabgarh - HSPCB | Ballabgarh | Haryana |
| HR004 | 76.141105 | 28.806223 | H.B. Colony, Bhiwani - HSPCB | Bhiwani | Haryana |
| HR005 | 76.7997 | 28.2068 | Municipal Corporation Office, Dharuhera - HSPCB | Dharuhera | Haryana |
| HR010 | 75.467934 | 29.503664 | Huda Sector, Fatehabad - HSPCB | Fatehabad | Haryana |
| HR015 | 75.744941 | 29.14056 | Urban Estate-II, Hisar - HSPCB | Hisar | Haryana |
| HR016 | 76.337619 | 29.307814 | Police Lines, Jind - HSPCB | Jind | Haryana |
| HR017 | 76.4155 | 29.8006 | Rishi Nagar, Kaithal - HSPCB | Kaithal | Haryana |
| HR018 | 77.0027 | 29.6953 | Sector-12, Karnal - HSPCB | Karnal | Haryana |
| HR019 | 76.875879 | 29.966942 | Sector-7, Kurukshetra - HSPCB | Kurukshetra | Haryana |
| HR020 | 76.9938 | 27.9002 | General Hospital, Mandikhera - HSPCB | Mandikhera | Haryana |
| HR021 | 76.93609 | 28.360699 | Sector-2 IMT, Manesar - HSPCB | Manesar | Haryana |
| HR022 | 75.730943 | 26.9502929 | Shastri Nagar, Narnaul - HSPCB | Narnaul | Haryana |
| KA001 | 75.659694 | 16.172806 | Vidyagiri, Bagalkot - KSPCB | Bagalkot | Karnataka |
| KA012 | 76.55521 | 11.55358 | Urban, Chamarajanagar - KSPCB | Chamarajanagar | Karnataka |
| KA013 | 77.731418 | 13.428828 | Chikkaballapur Rural, Chikkaballapur - KSPCB | Chikkaballapur | Karnataka |
| KA014 | 75.797056 | 13.328028 | Kalyana Nagara, Chikkamagaluru - KSPCB | Chikkamagaluru | Karnataka |
| KL001 | 76.302765 | 10.073232 | Udyogamandal, Eloor - Kerala PCB | Eloor | Kerala |
| MP002 | 79.446246 | 23.81748678 | Shrivastav Colony, Damoh - MPPCB | Damoh | Madhya Pradesh |
| MP003 | 76.064118 | 22.9682591 | Bhopal Chauraha, Dewas - MPPCB | Dewas | Madhya Pradesh |
| MP004 | 78.193251 | 26.203442 | City Center, Gwalior - MPPCB | Gwalior | Madhya Pradesh |
| MP006 | 75.5213 | 22.431 | Chhoti Gwaltoli, Indore - MPPCB | Indore | Madhya Pradesh |
| MP007 | 79.932247 | 23.168606 | Marhatal, Jabalpur - MPPCB | Jabalpur | Madhya Pradesh |
| MP008 | 80.23284 | 23.50016 | Gole Bazar, Katni - MPPCB | Katni | Madhya Pradesh |
| MH002 | 77.6345232 | 19.645324 | Chandrapur, Chandrapur - MPCB | Chandrapur | Maharashtra |
| MH004 | 73.142019 | 19.25292 | Khadakpada, Kalyan - MPCB | Kalyan | Maharashtra |
| PB002 | 74.907758 | 30.233011 | Hardev Nagar, Bathinda - PPCB | Bathinda | Punjab |
| PB003 | 75.578914 | 31.321907 | Civil Line, Jalandhar - PPCB | Jalandhar | Punjab |
| PB004 | 76.209694 | 30.736056 | Kalal Majra, Khanna - PPCB | Khanna | Punjab |
| PB005 | 75.8086 | 30.9028 | Punjab Agricultural University, Ludhiana - PPCB | Ludhiana | Punjab |
| PB006 | 76.331442 | 30.649961 | RIMT University, Mandi Gobindgarh - PPCB | Gobindgarh | Punjab |
| PB007 | 76.366642 | 30.349388 | Model Town, Patiala - PPCB | Patiala | Punjab |
| PB008 | 76.5623046 | 31.0325454 | Ratanpura, Rupnagar - Ambuja Cements | Rupnagar | Punjab |
| RJ001 | 76.611536 | 27.554793 | Moti Doongri, Alwar - RSPCB | Alwar | Rajasthan |
| RJ002 | 74.646594 | 26.470859 | Civil Lines, Ajmer - RSPCB | Ajmer | Rajasthan |
| RJ003 | 76.862296 | 28.194909 | RIICO Ind. Area III, Bhiwadi - RSPCB | Bhiwandi | Rajasthan |
| RJ007 | 84.9994 | 24.7955 | Collectorate, Jodhpur - RSPCB | Jodhpur | Rajasthan |
| RJ008 | 75.821256 | 25.14389 | Shrinath Puram, Kota - RSPCB | Kota | Rajasthan |
| RJ009 | 73.340227 | 25.771061 | Indira Colony Vistar, Pali - RSPCB | Pali | Rajasthan |
| RJ010 | 73.6321397 | 24.5886166 | Ashok Nagar, Udaipur - RSPCB | Udaipur | Rajasthan |
| UP003 | 77.849831 | 28.406963 | Yamunapuram, Bulandshahr - UPPCB | Bulandshahr | Uttar Pradesh |
| UP006 | 77.453839 | 28.685382 | Sanjay Nagar, Ghaziabad - UPPCB | Ghaziabad | Uttar Pradesh |
| UP008 | 77.482 | 28.47272 | Knowledge Park - III, Greater Noida - UPPCB | Greater Noida | Uttar Pradesh |
| UP010 | 77.749675 | 28.725645 | Anand Vihar, Hapur - UPPCB | Hapur | Uttar Pradesh |
| UP011 | 80.3229863 | 26.4703136 | Nehru Nagar, Kanpur - UPPCB | Kanpur | Uttar Pradesh |
| WB002 | 87.2892225 | 23.5404352 | Sidhu Kanhu Indoor Stadium, Durgapur - WBPCB | Durgapur | West Bengal |
| WB003 | 88.109737 | 22.06047 | Haldia, Haldia - WBPCB | Haldia | West Bengal |
| GJ002 | 73.010555 | 21.613267 | GIDC, Ankleshwar - GPCB | Ankleshwar | Gujarat |
| GJ004 | 73.010555 | 21.613267 | GIDC, Nandesari - Nandesari Ind. Association | Nandesari | Gujarat |
| GJ005 | 72.918013 | 20.362421 | Phase-1 GIDC, Vapi - GPCB | Vapi | Gujarat |
| HR023 | 77.3320667 | 28.1485564 | Shyam Nagar, Palwal - HSPCB | Palwal | Haryana |
| MH015 | 79.0517531 | 21.152875 | Opp GPO Civil Lines, Nagpur - MPCB | Nagpur | Maharashtra |
| MH016 | 73.7762427 | 20.0073285 | Gangapur Road, Nashik - MPCB | Nashik | Maharashtra |
| MH021 | 75.9063906 | 17.6599188 | Solapur, Solapur - MPCB | Solapur | Maharashtra |
| UP018 | 77.7622941 | 28.9535882 | Jai Bhim Nagar, Meerut - UPPCB | Meerut | Uttar Pradesh |
| UP019 | 77.709723 | 29.06351 | Pallavpuram Phase 2, Meerut - UPPCB | Meerut | Uttar Pradesh |
| UP021 | 77.7194031 | 29.4723508 | New Mandi, Muzaffarnagar - UPPCB | Muzzaffarnagar | Uttar Pradesh |
| UP022 | 77.3231257 | 28.5447608 | Sector - 125, Noida - UPPCB | Noida | Uttar Pradesh |
| KA015 | 75.140726 | 15.351773 | Deshpande Nagar, Hubballi - KSPCB | Hubballi | Karnataka |
| WB014 | 88.412668 | 26.6883049 | Ward-32 Bapupara, Siliguri - WBPCB | Siliguri | West Bengal |
| KA017 | 76.37376 | 12.21041 | Hebbal 1st Stage, Mysuru - KSPCB | Mysuru | Karnataka |
| KA018 | 77.298051 | 12.733409 | Vijay Nagar, Ramanagara - KSPCB | Ramanagara | Karnataka |
| MP010 | 77.511428 | 23.10844 | Sector-D Industrial Area, Mandideep - MPPCB | Mandideep | Madhya Pradesh |
| MP011 | 75.675238 | 22.624758 | Sector-2 Industrial Area, Pithampur - MPPCB | Pithampur | Madhya Pradesh |
| MP012 | 75.045981 | 23.331731 | Shasthri Nagar, Ratlam - IPCA Lab | Ratlam | Madhya Pradesh |
| UP025 | 77.393848 | 28.56923 | Sector-116, Noida - UPPCB | Noida | Uttar Pradesh |
| UP026 | 82.9083074 | 25.3505986 | Ardhali Bazar, Varanasi - UPPCB | Varanasi | Uttar Pradesh |
| KA016 | 76.822628 | 17.321993 | Lal Bahadur Shastri Nagar, Kalaburagi - KSPCB | Kalaburagi | Karnataka |

**Table S2.** Stations with PM$_{2.5}$ data but without NH$_3$ data

| Station | lon | lat | StationName | City | State |
|---|---|---|---|---|---|
| BR008 | 85.147382 | 25.619651 | Muradpur, Patna - BSPCB | Patna | Bihar |
| KL007 | 76.8865 | 8.5637 | Kariavattom, Thiruvananthapuram - Kerala PCB | Thiruvananthapuram | Kerala |
| MZ001 | 92.7192841 | 23.7176342 | Sikulpuikawn, Aizawl - Mizoram PCB | Aizawl | Mizoram |
| OD001 | 83.8396977 | 21.8004996 | GM Office, Brajrajnagar - OSPCB | Brajrajnagar | Odisha |
| OD002 | 85.1707021 | 20.9360711 | Talcher Coalfields, Talcher - OSPCB | Talcher | Odisha |

**Table S3.** Stations with $NH_3$ data but without $PM_{2.5}$ data

*RC: 27) Figure 2: Row (b) seems useless and not mentioned in the text, consider moving to the supplementary.*

**AR**: Thanks for comment. I have moved row (b) of Figure 2 to the supplementary material, as it is not discussed in the main text.

***Figure in supplement:***

[Figure]

Figure S1. The distribution of IASI instruments' uncertainty in 2019 January (a), April (b), July (c) and November (d).

*RC: 28) Figure 2: The way you plotted the IASI and GC seems different.*

**AR**: Many thanks for your feedback. Yes, we use scatter plots for the IASI data, while the model simulations are presented using filled contour maps to show their respective spatial distributions.

***Figure in manuscript:***

[Figure]

Figure 1. Spatial distribution of the total column $NH_3$ concentration from IASI (a) or CrIS (b) instruments, and from the GEOS-Chem simulation either using the prior (c) or using the posterior (d) $NH_3$ emission felx in 2019 January (a.1)–(d.1), April (a.2)–(d.2), July (a.3)-(d.3) and November (a.4)–(d.4).

*RC: 29) Figure 3: I am curious why there are 0 values (grey area) in the CEDS prior and posterior, and how you deal with this 0 values or missing gap in the posterior. It may look confused since there are lots of grey area in (a) and (b) but not shown in (c)*

**AR**: Thanks for comment. Apologies for the confusion. The values in the grey area are not zero, but rather fall within the range of 0 to 0.2. They are indeed non-zero, but the values are quite small. We have treated these small values as valid and have not excluded them in the posterior, which may explain why they appear differently in panel (c).

*Text in manuscript:*

[Figure]

Figure 3. Spatial distribution of the prior (a), the posterior (b) and the posterior minus prior increments (c) monthly NH₃ emission in 2019 January (a.1)–(c.1), April (a.2)–(c.2) , July (a.3)-(c.3) and November (a.4)–(c.4).

*RC: 30) NH₃ total column concentration: "Four months": in 2019. Note all month are presented in Fig.4.*

**AR**: Apologies for the confusion, we have revised this statement in the original text. The specific revised fragment is as follows:

*Text in manuscript:*

*3.1.1 Spatial* NH₃ *total column concentration*

The prior and posterior snapshot of NH₃ column concentration simulations for four months (January, April, July, and November) are presented in Fig. 1 (c)-(d), alongside the IASI measurements shown in panel (a).

*RC: 31) "Distributions for the rest months": could you provide timeseries or seasonal pattern of these two satellite observations?*

**AR**: Thanks for comment. The time series for the two satellites are presented in panel (a) of Figure

4, while the spatial distribution is shown in Figure S3.

***Figure in manuscript and supplement:***

[Figure]

Figure 4. The monthly average total NH₃ column concentrations from the prior and posterior IASI-observed, and CrIS-observed fromJanuary to December (a). The monthly average values of prior and posterior emissions from January to December (b).

[Figure]

Figure S3. The distribution of the IASI-observed (a) and the CrIS-observed (b) ammonia total column for the remain months in 2019.

*RC: 32) "background error covariance matrix": does this refer to B in Eq.3 and 5?*

**AR**: Yes, the "background error covariance matrix" refers to **B** in Equations 3 and 5.

*Text in manuscript:*
* * *
*2.3 Emission inversion system*

…

$$\mathcal{J}(\boldsymbol{f}) = \frac{1}{2}(\boldsymbol{f} - \boldsymbol{f}_b)^{\mathrm{T}} \mathbf{B}^{-1}(\boldsymbol{f} - \boldsymbol{f}_b) + \frac{1}{2}\{\boldsymbol{y} - \mathbf{H}\mathcal{M}(\boldsymbol{f})\}^{\mathrm{T}} \mathbf{O}^{-1}\{\boldsymbol{y} - \mathbf{H}\mathcal{M}(\boldsymbol{f})\} \tag{3}$$

$$\mathbf{B}(i,j) = \sigma^2 \cdot \boldsymbol{f}(i) \cdot \boldsymbol{f}(j) \cdot \mathbf{C}(i,j) \tag{5}$$

*3.1.1    Spatial NH₃ total column concentration*

…

Note that there are still some discrepancies in the posterior simulation vs IASI column measurements. In particular, as shown in panel a.3 vs. d.3 of Fig. 1, the posterior simulation did not fully reproduce the extremely high $NH_3$ loading observed by IASI in July (with column-integrated concentrations exceeding $10 \times 10^{16}$ molec cm$^{-2}$). This occurs because the goal of the assimilation is to achieve the best fit between the posterior, the observations, and the prior emissions, rather than just fitting the observations alone. The extremely high $NH_3$ concentrations are less likely given the relatively low prior $NH_3$ emissions and the background error covariance matrix described in Section 2.3.

*RC: 33) $NH_3$ and $PM_{2.5}$ ground concentration validation: "The mismatch": specify which kind of mismatch, bias mismatch or spatial/temporal*

**AR**: Thanks for comment. The mismatch mentioned refers to both bias and spatial mismatches. The bias mismatch is observed in the differences between the ground observations and simulations, where ground $NH_3$ concentrations were occasionally higher than the simulations. The spatial mismatch could be due to the locations of the monitoring stations, which are often in urban areas where $NH_3$ levels are higher due to traffic and human activities.

*RC: 34) Can also add the Bias in Fig. 4, 5 and 7.*

**AR**: Many thanks for your feedback. I have added the bias to Figures 4, 5, and 7 as requested.

*RC: 35) Fig. 4 and 5 can be combined.*

**AR**: Thanks for comment. I have combined Figures 4 and 5 as per your recommendation. The revised figure along with RC (34) is as follows:

*Figure in manuscript:*

[Figure]

Figure 6. Scatter plot of the IASI (a-b) and CrIS (c-d) observed NH₃ concentrations against the NH₃ simulation over South Asia, either using the prior or the posterior NH₃ emission inventory, from January to December.

[Figure]

Figure 7. Scatter plot of the ground-observed against the NH₃ simulation over South Asia either using the prior (a) or using the posterior (b) NH₃ emission inventory in 2019.

*RC: 36) Fig. 2 and 6 can be combined.*

**AR**: Thanks for comment. I have combined Figures 2 and 6 as per your recommendation. The revised figure is as follows:

*Figure in manuscript:*

[Figure]

Figure 1. Spatial distribution of the total column $NH_3$ concentration from IASI (a) or CrIS (b) instruments, and from the GEOS-Chem simulation either using the prior (c) or using the posterior (d) $NH_3$ emission flex in 2019 January (a.1)–(d.1), April (a.2)–(d.2) , July (a.3)- (d.3) and November (a.4)–(d.4).

*RC: 37) "systematic biases": how do you define the systematic biases, which differentiate from the random error?*

**AR**: Thanks for comment. Systematic bias is a consistent and predictable deviation, usually caused by factors such as biases in input data (e.g., emission inventories). This type of error typically leads to the model consistently overestimating or underestimating the actual values in its predictions, whereas random errors are unpredictable and irregular, occurring due to factors that are not consistent across different measurements or observations.

*RC: 38) Seasonal and annual variation of ammonia concentration: The peaks in* $NH_3$ *posterior concentration (May) and emission (July) are different and interesting, can you elaborate to explain it?*

**Reply**: Thank you for your kind comment. Although the posterior emission shows that July has the highest emissions, this emission is mainly due to human activities. Specifically, biomass combustion emissions are highest in spring, especially in May in Figure 4 (c), and anthropology emissions are also one of the peaks. Although anthropology emissions are highest in July, meteorological factors must also be considered. As shown in the Figure 4 (a) and (b), precipitation and temperature in July are the highest of the year. High temperatures increase ammonia volatilization, and the high precipitation increases the wet deposition of ammonia. These combined factors lead to July having a smaller concentration peak compared to May, despite being another peak month. The revised related discussion in the text is as follows:

*Text in manuscript:*

3.1.2 *Seasonal and annual variation of* $NH_3$ *concentration*

…

The high value in May is attributed to huge amount of biomass burning in South Asia during the spring in Figure S4 (c). We have identified the planting and harvesting times of crops in the South Asia region from USDA(U.S.DEPARTMENT OF ARGRICULTURE, https://ipad.fas.usda.gov/rssiws/al/crop_calendar/sasia.aspx). The heavy use of fertilizers in agricultural activities has resulted in the highest emission throughout the year, as will be illustrated in Fig. 4 (b) in Section 3.2. This has lead to the second $NH_3$ concentration peak in July. The reasons for higher emissions in July but lower concentration levels compared to May could be attributed to meteorological factors. The monsoon season in South Asia results in increased wet deposition, and notably, 2019 experienced the most intense monsoon since 1994 (NASA, 2020). As shown in the Figure S4 (a) and (b), precipitation and temperature in July are the highest of the year. High temperatures increase ammonia volatilization, and the high precipitation increases the wet deposition of ammonia. These combined factors lead to July having a smaller concentration peak compared to May, despite being another peak month.

…

3.2 *Spatial and Seasonal variation of* $NH_3$ *emission*

…

The substantial emissions in July, as indicated by the posterior inventory, can be attributed to the increased fertilizer application for rice and corn crops during the summer season (Tanvir et al., 2019). Although biomass burning emissions are generally higher in spring in Figure S4 (c), agricultural activities remain the primary contributors to $NH_3$ emissions (Huang et al., 2016),

resulting in July surpassing May in emission intensity. From July to September, as rice and other crops progress through their growth stages, fertilizer application typically decreases, leading to a gradual reduction in $NH_3$ emissions. Additionally, temperatures decline from August to September in Figure S4 (b), reducing the volatilization rate of $NH_3$. This pattern occurs because $NH_3$ volatilization is strongly influenced by temperature (Fan et al., 2011)

[Figure]

Figure 4. The monthly average total $NH_3$ column concentrations from the prior and posterior, IASI-observed, and CrIS-observed from January to December (a). The monthly average values of prior and posterior emissions from January to December (b).

***Supplement***

[Figure]

Figure S4. Monthly precipitation (a), temperature (b) from MERRA2 and biomass buring emission (c) from GFED4 in 2019.

*RC: 39) Spatial and Seasonal variation of ammonia emission: No posterior data in Fig.S3.*

**AR**: Thanks for comment. We have now added the posterior data to Figure S3 as requested.

.

*Figure in manuscript:*

[Figure]

Figure S5. (a)-(l) represent the distribution of the posterior inventory for each month from January to December in 2019.

---

## Author Comment (AC2)

**Authors' Response to Reviews of**

**South Asia ammonia emission inversion through assimilating IASI observations**

Ji Xia[*], Yi Zhou[*], Li Fang, Yingfei Qi, Dehao Li, Hong Liao, and Jianbing Jin
*Atmospheric Chemistry and Physics Discussions,*
* * *
RC: Reviewers' Comment, AR: Authors' Response, □ Manuscript Text

**General Comments:**

**RC:** *The paper describes the application of an ammonia emission inversion system over South Asia. As emissions of ammonia are rather difficult to estimate by emission models, this approach is very useful to obtain insight in the actual emission strength. Especially the time period of the seasonal emission peak(s) is difficult to model, but the described system seems able to provide a better estimate for that. The system uses IASI satellite observations to constrains the emissions; results are validated by comparing posterior simulations with observations from the CrIS satellite instrument and observations from a ground network. The temporal resolution of the emission estimates is monthly, which is rather course compared to the high frequency changes present in ammonia emissions. For the described study that seems a logical choice, as also the prior emission inventory is monthly.*
*Although the paper focusses on the application of the inversion system, the setup of the inversion is described sufficiently well. For some parts a more detailed description could be useful, as described below in the Specific Comments. Overall, the paper is easy to read, and could be published after some minor modifications.*

**Response to Referee #2**: We would like to thank the referee for the careful review throughout the paper and the in-depth comments that help to improve our paper.

**Specific Comments:**

**RC:1) could the authors discuss the potential of their system for higher temporal resolution estimates? What are the current limitations for application on weekly or even daily scale? Is the availability and/or quality of the satellite observations a limitation, or simply the computing resources?**

**AR**: Thanks for comment. The system already has the potential for higher temporal resolution estimates. With data available from the FY-4 satellite, we can access daily or even two-hourly observations. These datasets allow us to explore higher temporal resolution (e.g., daily or weekly

scale) spatiotemporal variation characteristics. Our next steps will focus on further refining the spatiotemporal patterns at the daily or weekly scale, building on the current posterior results.

As for the limitations, the primary constraint is not the availability or quality of satellite observations, but rather the computational resources required. While satellite data is sufficient to support higher temporal resolution estimates, processing these datasets at high spatial and temporal resolutions demands significant computational power and storage. Moving forward, we will optimize the computational methods and leverage more powerful computational platforms to achieve higher resolution temporal estimates.

*Text in manuscript*:
* * *
*4 Summary and conclusion*

*…*

The top-down $NH_3$ emission inversion system driven by IASI observations has demonstrated superior performance in enhancing the $NH_3$ emission estimates. Nevertheless, several challenges persist, such as the requirement for simulations at finer resolutions to precisely capture very local emission dynamics. Furthermore, observations from stationary satellites, such as FY-4B, also deserve attention for exploring the diurnal variations of the $NH_3$ emission. Our next steps will focus on further refining the spatiotemporal patterns at the daily or weekly scale, building on the current inversion system.
* * *
*RC: 2) The inversion now uses IASI observations to constrain emissions, and CrIS observations for validation. Would it be possible to use instead the CrIS observations to constrain the emissions? The results show quite some differences between IASI and CrIS $NH_3$ columns; would inversion of CrIS data give very different results?*

**AR**: Appreciate your comment. The results of emission inversions are highly dependent on the type of observations used. Different sensors provide different sensitivities and spatial coverage, which can lead to variations in the inversion outcomes. We chose to use IASI observations for the inversion because IASI provides a larger volume of data, which increases the robustness of the emission estimates. Since IASI observations are collected from three satellites, this results in an even larger dataset. While CrIS observations are valuable, we did not use them in this study because we have indicated underestimation issues in South Asia region with CrIS $NH_3$ columns. At the surface, where CrIS typically has lower sensitivity, it tends to overestimate in low-concentration conditions and underestimate in higher atmospheric concentration conditions (Dammers E et al., 2017). As a result, we opted for IASI data, which provides more reliable constraints in this case. Additionally, the posterior emission estimates, which are based on CrIS, have now been included as supplementary material. Furthermore, we have also added this point in the manuscript to ensure clarity.

*Text in manuscript:*
* * *
*3.2 Spatial and Seasonal variation of NH3 emission*

*...*

The convergence of prior and posterior emission intensities in June is attributed to the overall offsetting of negative and positive increments in the region, as shown in Figure S7 (f). As depicted in panel (c) of Fig. 3, the negative increments observed in January and April primarily originate from the Indian region, while the positive increments in July and September are predominantly observed in the same area. Additionally, the posterior emission estimates, which are based on CrIS, have now been included as supplementary material.
* * *
*RC: 3) p 3, line 32: When the monthly averages over grid cells are calculated, is there any spatial or temporal weighting applied? For example, a spatial weight based on the overlap between a pixel footprint and the target grid cell, or a temporal weight based on the instrument error?*

**AR**: Many thanks for your feedback. We did not apply any spatial or temporal weighting. The data is processed by reading daily $NH_3$ values and adding them to the corresponding grid cells. For each valid data point, if its latitude and longitude fall within the specified range, its $NH_3$ value is added to the corresponding grid cell.

*RC: 4) p 4, line 14-15: What are the units of these variables? A more standard formulation of the kernel application would look like:*

$$Xm = Xa + A(m - B)$$

*Could Eq (1) be rewritten to this actually?*

*How is the averaging kernel applied to the model data exactly? Is a monthly averaged kernel applied to monthly averaged concentration? If so, how is the monthly averaged model concentration calculated, as an average over all ours, or using time of overpass only? Or are the individual pixels simulated from the model first, and then averaged over grid cells and months?*

**AR**: Thank you for your valuable comment. Apologies for the confusion. There are two methods in (Clarisse et al., 2023) and we used the model vertical profile as a prior to recalculate the ammonia column concentration from IASI. So the formula for this method is still different from Xm = Xa + A (m - B). To clarify, in Eq. 1:

$$m_z = \frac{M_z^m - B_z}{M^m - B}$$

$M_z^m$ represents the modeled concentration of NH₃ at altitude **z**.

$B_z$ is the background concentration of NH₃ at the same altitude.

$M^m$ represents the total modeled concentration of NH₃ in the atmosphere.

**B** is the total background concentration.

$$A_z^a = \frac{1}{N}\frac{\hat{X}^a - B}{\hat{X}^{|z} - B}$$

$\hat{X}^{|z}$ represents the a priori (or assumed) concentration of NH₃ at altitude **z**.

$B_z$ is again the background concentration at that altitude.

$\hat{X}^a$ is the total a priori concentration.

**N** is a normalization factor, ensuring the matrix $A_z^a$ sums correctly to account for all altitudes.

The specific description of the relevant part is as follows:

*Text in manuscript:*
* * *
**2.1 *IASI satellite measurements***

…

Here, $\hat{X}^m$ represents the IASI column concentration retrieved with model profile. $\hat{X}^a$ denotes the initial IASI column concentration, with the background concentration B. The Aa z values are AVK for each vertical layer, with the model profile $m_z$. More detailed information and the corresponding equations are provided in the supplementary materials equation S8 and S9.

***Supplement***

…

$$m_z = \frac{M_z^m - B_z}{M^m - B}$$

here $M_z^m$ represents the modeled concentration of NH₃ at altitude z. $B_z$ is the background concentration of NH₃ at the same altitude. $M^m$ represents the total modeled concentration of NH₃ in the atmosphere. B is the total background concentration.

$$A_z^a = \frac{1}{N}\frac{\hat{X}^a - B}{\hat{X}^{|z} - B_z}$$

here $\hat{X}^{|z}$ represents the a priori (or assumed) concentration of NH₃ at altitude z. $B_z$ is again the background concentration at that altitude. $\hat{X}^a$ is the total a priori concentration. N is a normalization factor, ensuring the matrix $A_z^a$ sums correctly to account for all altitudes.
* * *
*RC: 5) Would it also be possible to not use monthly averaged observations, but simply all observations individually? The estimated emission state could still be monthly, so what is the reason for using monthly averaged observations?*

**AR**: Thanks for comment. Using all individual observations without averaging would indeed be

possible, but there are two main reasons we use monthly and grid averaging. First, averaging aligns the satellite observations with the model simulation's grid resolution, ensuring that we are comparing like with like. Second, averaging dramatically reduces the size of the observational vector used in the assimilation, which in turn lowers the computational cost. For example, without averaging, the observational vector y might have a size of around 1,000,000, whereas with monthly and grid averaging, it is reduced to about 1,000. This reduction makes the assimilation analysis far more computationally efficient while still accurately representing the monthly emission state. We have also added relevant explanations for this section in the article belew:

*Text in manuscript*:
* * *
**2.1 IASI satellite measurements**

…

The assimilated observations for estimating the $NH_3$ emissions were the monthly IASI column concentration means over the 0.5 °× 0.625 °GEOS-Chem grid cell. These values were derived from the latest ANNI-$NH_3$-v4R-ERA5 product. Despite improvements in $NH_3$ column retrievals from satellite observations, there remains substantial variability in measurement uncertainty, ranging from 5 % to over 1000 %. (Van Damme et al., 2014; Whitburn et al., 2016; Van Damme et al., 2017). Data selection was performed by excluding nighttime observations, irrational values (<0), and only using data with a cloud fraction< 0.1 (Van Damme et al., 2018) and skin temperature > 263 K (Van Damme et al., 2014) during the calculation of the monthly mean. Additionally, while negative values are not necessarily incorrect, they are considered unrealistic in the context of $NH_3$ concentrations. To improve the quality of the monthly average, we removed those negative values. It is also important to note that we used daily observations from three satellites, each with a pixel resolution of approximately 12 km × 12 km, which provided us with sufficient observations to calculate the monthly average. We applied a selection criterion, using only grid averages that contain a minimum of 80 observations. This ensures that the grid-averaged values are statistically representative and that the monthly mean is of high quality. Notably, the time coverage of the available version 4 IASI product used was limited: Metop-A provided data for the entire year of 2019, Metop-B provided data from January to July 2019, and Metop-C did not have data for 2019. Therefore, only the data from Metop-A and Metop-B within the 2019 time frame were used in this study. To further improve the data quality and ensure consistency, we performed monthly and grid averaging of the observations. This approach not only allows for a fair comparison between the observed and modeled $NH_3$ concentrations but also reduces the computational cost of the assimilation process. Using individual observations without averaging would result in an excessively large observational vector, which would significantly increase the computational burden. For example, without

averaging, the size of the observational vector could reach 1,000,000, while with monthly and grid averaging, it is reduced to a manageable size of around 1,000. This reduction in size helps to optimize the data assimilation process while maintaining the integrity of the emission estimates.

**RC: 6) Negative values are not necessarily wrong. The uncertainty of these values is probably high, so the "true" value is still a very likely outcome. By removing the negative observations, the monthly average will have a positive bias. Could this be discussed?**

**AR**: Thank you for your comment. We acknowledge that negative values are not necessarily wrong in an assimilation context if they have large uncertainties. However, in a physical sense, negative $NH_3$ concentrations are not realistic. To improve the quality of the monthly averages, we removed these negative values. It is also important to note that our monthly averages are calculated using daily observations from three satellites, each with a pixel resolution of 12km $\times$ 12 km. This means we have a large number of observations available for each grid cell. Additionally, we apply a selection criterion: a grid average is only used if it contains a minimum of 80 observations. This ensures that the final grid-averaged value is statistically representative and minimizes potential biases in the monthly mean. Given this approach, we believe that removing negative values does not introduce a significant positive bias but rather enhances the reliability of the data used in the assimilation. We have also added relevant explanations for this section in the article below:

*Text in manuscript:*

*2.1 IASI satellite measurements*

*…*

The assimilated observations for estimating the $NH_3$ emissions were the monthly IASI column concentration means over the 0.5 °× 0.625 °GEOS-Chem grid cell. These values were derived from the latest ANNI-$NH_3$-v4R-ERA5 product. Despite improvements in $NH_3$ column retrievals from satellite observations, there remains substantial variability in measurement uncertainty, ranging from 5 % to over 1000 %. (Van Damme et al., 2014; Whitburn et al., 2016; Van Damme et al., 2017). Data selection was performed by excluding nighttime observations, irrational values (<0), and only using data with a cloud fraction< 0.1 (Van Damme et al., 2018) and skin temperature > 263 K (Van Damme et al., 2014) during the calculation of the monthly mean. Additionally, while negative values are not necessarily incorrect, they are considered unrealistic in the context of $NH_3$ concentrations. To improve the quality of the monthly average, we removed those negative values. It is also important to note that we used daily observations from three satellites, each with a pixel resolution of approximately 12 km $\times$ 12 km, which provided us with sufficient observations to calculate the

monthly average. We applied a selection criterion, using only grid averages that contain a minimum of 80 observations. This ensures that the grid-averaged values are statistically representative and that the monthly mean is of high quality. Notably, the time coverage of the available version 4 IASI product used was limited: Metop-A provided data for the entire year of 2019, Metop-B provided data from January to July 2019, and Metop-C did not have data for 2019. Therefore, only the data from Metop-A and Metop-B within the 2019 time frame were used in this study. To further improve the data quality and ensure consistency, we performed monthly and grid averaging of the observations. This approach not only allows for a fair comparison between the observed and modeled $NH_3$ concentrations but also reduces the computational cost of the assimilation process. Using individual observations without averaging would result in an excessively large observational vector, which would significantly increase the computational burden. For example, without averaging, the size of the observational vector could reach 1,000,000, while with monthly and grid averaging, it is reduced to a manageable size of around 1,000. This reduction in size helps to optimize the data assimilation process while maintaining the integrity of the emission estimates.

*RC: 7) p6, lines 8-10: Why is this minimum value chosen, how often does it have this value? Are the gray values in Figure 2 "b" this minimum? Maybe better to move this part to Section 2.1 where the observation uncertainty is discussed.*

**AR**: Appreciate your comment. The minimum value used for the assimilation process is empirically chosen to avoid overemphasizing extremely low measurements that are likely unreliable due to high uncertainty. This value is rarely used and occurs with very low frequency, approximately 3% of the observations. We also note that the gray values in Figure 2(b) represent the uncertainty in the observations, not the minimum value used in the selection process. However, we have kept this part in the current section because Section 2.1 primarily focuses on the processing of satellite observations and their associated uncertainties, which is distinct from the inversion system discussed here. The details of the observation error covariance matrix (O) and the related formula are directly relevant to the emission inversion process, which occurs in this section. Therefore, we believe that placing this explanation in the current context provides a clearer understanding of how the uncertainties are incorporated into the inversion system.

*RC: 8) p 14, section 3.2: Fig 9.a shows prior and posterior model columns, are these after application of averaging kernels? Then the lines should be different for IASI and CrIS. If these are model columns, how well could these be compared to the satellite columns?*

**AR**: Thank you for your comment. As answered above regarding the use of averaging kernels, model column concentrations are not processed with averaging kernels, while satellite column concentrations are corrected using averaging kernels to align them more closely with the vertical distribution of the model. After correction, the satellite column concentrations are adjusted to match

the model's three-dimensional structure, ensuring that they have a similar physical basis before comparison. We have also strengthened some of the discussion regarding this figure in the article, as follows:
* * *
*Text in manuscript*

3.1.2 *Seasonal and annual variation of* $NH_3$ *concentration*

...

The high value in May is attributed to huge amount of biomass burning in South Asia during the spring in Figure S4 (c). We have identified the planting and harvesting times of crops in the South Asia region from USDA(U.S.DEPARTMENT OF ARGRICULTURE, https://ipad.fas.usda.gov/rssiws/al/crop_calendar/sasia.aspx). The heavy use of fertilizers in agricultural activities has resulted in the highest emission throughout the year, as will be illustrated in Fig. 4 (b) in Section 3.2. This has lead to the second $NH_3$ concentration peak in July. The reasons for higher emissions in July but lower concentration levels compared to May could be attributed to meteorological factors. The monsoon season in South Asia results in increased wet deposition, and notably, 2019 experienced the most intense monsoon since 1994 (NASA, 2020). As shown in the Figure S4 (a) and (b), precipitation and temperature in July are the highest of the year. High temperatures increase ammonia volatilization, and the high precipitation increases the wet deposition of ammonia. These combined factors lead to July having a smaller concentration peak compared to May, despite being another peak month.

...

3.2 *Spatial and Seasonal variation of* $NH_3$ *emission*

...

The substantial emissions in July, as indicated by the posterior inventory, can be attributed to the increased fertilizer application for rice and corn crops during the summer season (Tanvir et al., 2019). Although biomass burning emissions are generally higher in spring in Figure S4 (c), agricultural activities remain the primary contributors to $NH_3$ emissions (Huang et al., 2016), resulting in July surpassing May in emission intensity. From July to September, as rice and other crops progress through their growth stages, fertilizer application typically decreases, leading to a gradual reduction in $NH_3$ emissions. Additionally, temperatures decline from August to September in Figure S4 (b), reducing the volatilization rate of $NH_3$. This pattern occurs because $NH_3$ volatilization is strongly influenced by temperature (Fan et al., 2011)

[Figure]

Figure 4. The monthly average total NH$_3$ column concentrations from the prior and posterior, IASI-observed, and CrIS-observed from January to December (a). The monthly average values of prior and posterior emissions from January to December (b).

**Supplement**

[Figure]

Figure S4. Monthly precipitation (a), temperature (b) from MERRA2 and biomass buring emission (c) from GFED4 in 2019.

*RC: 9) p 2, line 13: "compared emissions of other pollutants"*

**AR**: Thanks for comment. I have revised the sentence to clarify the "comparison of emissions of other pollutants". The following is the revised abstract excerpt:

*Text in manuscript:*

Over the past decade, scientists have predominantly employed the "bottom-up" approach to estimate NH$_3$ emissions. When combined with chemical transport models, atmospheric NH$_3$ dynamics can be simulated, enabling the quantification of environmental impacts. Substantial efforts have been made to quantify the spatiotemporal distribution of NH$_3$ sources and develop global/regional emission inventories, such as the global NH$_3$ emission inventory (Bouwman et al., 1997), the anthropogenic emission inventory that includes NH$_3$ estimates (e.g., Community Emissions Data System, CEDS) (Hoesly et al., 2018), as well as regional NH$_3$ inventories focusing on South Asia (Yan et al., 2003; Yamaji et al., 2004; Liu et al., 2022). However, these bottom-up estimates of NH$_3$ emissions are generally considered as uncertain (Xu et al., 2019), particularly when compared emissions of other pollutants primarily originating from fossil fuel combustion such as NO2.

*RC: 10) p 2, line 2: Add a reference here?*

**AR**: Thanks for comment. We have added a reference here and also updated the literature related to

climate effects to make it more relevant and up-to-date. The following is the revised abstract excerpt:

*Text in manuscript:*

> Further, ammonia gas, along with its reaction products, plays a pivotal role in soil acidification and the eutrophication of water bodies through both dry and wet deposition (Krupa, 2003), and thereby affecting the balance of ecosystems (Asman et al.,1998) and climate change (Ma et al., 2022; Gong et al., 2024).

*RC:11) p 4, line 16: The uncertainty assigned to the IASI measurements is also an essential"*

**AR**: Many thanks for your feedback. We have made the revision accordingly. The following is the revised abstract excerpt:

*Text in manuscript:*

> *1.1 IASI satellite measurements*
>
> *…*
>
> The uncertainty assigned to the IASI measurements is also an essential. When calculating the uncertainty of gridded monthly average $NH_3$ measurements, both instrumental errors $\sigma^{instrumental}$ and representation error $\sigma^{representation}$ are considered. The gridded average uncertainty derived directly from IASI products was designated as instrumental error $\sigma^{instrumental}$, while the standard deviation of the observed samples for the gridded average characterized representation error $\sigma^{representation}$.

*RC: 12) p4 line 24: Fig 2 is referenced before Fig 1, change order of figures?*

**AR**: Thanks for comment. We have adjusted the order of the figures accordingly. Below is the revised order of the figures.

*Text in manuscript:*

[Figure]

Figure 1. Spatial distribution of the total column NH$_3$ concentration from IASI (a) or CrIS (b) instruments, and from the GEOS-Chem simulation either using the prior (c) or using the posterior (d) NH$_3$ emission flex in 2019 January (a.1)–(d.1), April (a.2)–(d.2) , July (a.3)-(d.3) and November (a.4)–(d.4).

[Figure]

Figure 2. The GEOS-Chem model simulation domain, with dots indicating the locations of ground observation stations from the Central Pollution Control Board (CPCB), India. The three different colored dots represent stations with only PM2.5 observations, stations with both PM2.5 and NH$_3$ observations, and stations with only NH$_3$ observations, respectively.

*RC: 13) p 5, line 22: Explain that the emission field is "f"; what are the units?*

**AR**: Thank you for your kind comment. We have added a detailed description of this parameter in the manuscript. The relevant information from the article is shown as follows:

*Text in manuscript:*

> *2.3Emission inversion system*
>
> …
>
> Here, $f$ denotes the vector of the NH$_3$ estimated emission field, with its units typically expressed in kg/m2/s.

*RC: 14) p 6, line 3: Shouldn't this be fb?*

**AR**: Appreciate your comment. Yes, this should be $f_b$. We have corrected the formula (5) in the manuscript accordingly.

*Text in manuscript:*

> *2.3Emission inversion system*
>
> …
>
> $$\mathbf{B}(i,j) = \sigma^2 \cdot \boldsymbol{f_b}(i) \cdot \boldsymbol{f_b}(j) \cdot \mathbf{C}(i,j)$$

*RC: 15) p 6, line 7: Shouldn't this be "observation representation errors are independent from each other"?*

**AR**: Thanks for comment. I updated this sentence to state "observation representation errors are independent from each other." The following is the revised abstract excerpt:

*Text in manuscript:*

> *2.3 Emission inversion system*
>
> …
>
> while $\mathbf{O}$ is the observation error covariance matrix. Here we assume IASI observation representation errors are independent from each other.

*RC:16) p6 line 8: "as described in"*

**AR**: Thanks for comment. I added "as described in" for clarity. The following is the revised abstract

excerpt:

*Text in manuscript:*

> **2.3 Emission inversion system**
>
> …
>
> while **O** is the observation error covariance matrix. Here we assume IASI observation representation errors are independent from each other. O therefore is a diagonal matrix filled with the square of the integrated uncertainty as described in Section 2.1.

*RC: 17) p8, line 19: "as well as manure from livestock, including cattle, ..."*

**AR**: Appreciate your comment. I revised this line to include the mention of manure from livestock, including cattle. The following is the revised abstract excerpt:

*Text in manuscript:*

> **2.4 GEOS-Chem model and emission inventory**
>
> The NH$_3$ emissions inventory employed to drive GEOS-Chem originated from the Community Emissions Data System (CEDS, https://doi.org/10.25584/PNNLDH/1854347) inventory, which was widely used for modeling the South Asia atmo-spheric pollutants, e.g., VOCs (Chaliyakunnel et al., 2019), PM2.5 pollution (Guttikunda and Nishadh, 2022; Xue et al., 2021). CEDS inventory includes various sources encompassing agricultural, energy production, industrial, residential and commercial activities, ships, solvent use, surface transportation, and waste processing (McDuffie et al., 2020), the bulk of NH$_3$ emissions originate from agricultural practices. Specifically, these emissions stem predominantly from farmlands, including crops such as wheat, maize, and rice, as well as manure from livestock, including cattle, chicken, goats, and pigs (Liu et al., 2022).

*RC: 18) p 8, line 25: "posterior result"*

**AR**: I updated "posterior result" as requested. At the same time, we have reorganized the logical structure of this section, and the revised fragment is as follows:

*Text in manuscript:*

> With the assimilation system described above, the monthly NH$_3$ emission inversion for 2019 over South Asia is conducted. The Spatial of prior and posterior results are in Section 3.1.1. The long-term varying trend of South Asia NH$_3$ emission is illustrated in Section 3.1.2, followed by an analysis and discussion of its spatial distribution and seasonal profile based on the inversion results in Section 3.2. Then the posterior result is evaluated in Section 3.3.

*RC: 19) p 8, lines 26-27: mention 3.2 first, then 3.3*

**AR**: Many thanks for your feedback. I reordered the references to sections 3.2 and 3.3 as per your suggestion.

We have also revised the structure of the Results and Discussion section. The updated structure is as follows:

*Text in manuscript:*
* * *
*3 Results and discussion*

*3.1 Observed* NH$_3$ *concentrations*

*3.1.1 Spatial* NH$_3$ *total column concentration*

*3.1.2 Seasonal and annual variation of* NH$_3$ *concentration*

*3.2 Spatial and Seasonal variation of* NH$_3$ *emission*

*3.3 Validation*

*3.3.1* NH$_3$ *total column concentration validation*

*3.3.2* NH$_3$ *and PM2.5 ground concentration validation*
* * *
*RC: 19) p 16, Figure 10 caption: (j) is a time series, not a scatter plot. And the box plots represent yearly averages.*

**AR**: Thanks for comment. I changed the description of panel (j) to indicate that it is a time series, not a scatter plot. I also updated the caption to explain that the box plots represent yearly averages. The following is the revised abstract excerpt:

*Text in manuscript:*

[Figure]

Figure 5. The spatial distribution of the annual averaged IASI column concentrations in South Asia from 2015 to 2023 is shown in panels (a) to (i). Panel (j) presents a time series depicting the monthly variation in IASI-observed $NH_3$ column concentrations from 2015 to 2023, with the box plots representing the yearly averages showing interannual changes.

**Reference**

Dammers E, et al. "Validation of the CrIS fast physical NH 3 retrieval with ground-based FTIR。" *Atmospheric Measurement Techniques Discussions, 2017(2017): 1-32.*

Ma, Ruoya, et al. "Data-driven estimates of fertilizer-induced soil $NH_3$, NO and $N_2O$ emissions from croplands in China and their climate change impacts." *Global Change Biology 28.3 (2022): 1008-1022.*

Gong, Cheng, et al. "Global net climate effects of anthropogenic reactive nitrogen." *Nature 632.8025 (2024): 557-563.*

---

## Author Response (AR2)

**Authors' Response to Reviews of**

**South Asia anthropogenic ammonia emission inversion through assimilating IASI observations**

Ji Xia*, Yi Zhou*, Li Fang, Yingfei Qi, Dehao Li, Hong Liao, and Jianbing Jin
*Atmospheric Chemistry and Physics Discussions,*
* * *
RC: Reviewers' Comment, AR: Authors' Response, □ Manuscript Text

**General Comments:**

RC: *The authors have addressed most of my previous comments effectively, with a more logical structure and clearer figures. However, the following minor revisions are still required to strengthen the ms:*

Response to Referee #1: We would like to thank the referee for the careful review throughout the paper and the in-depth comments that help to improve our paper.

**Specific Comments:**

RC: *1) Units consistency: $NH_3$ concentration units are inconsistently reported in the Introduction and Results sections. To improve readability, use uniform units throughout the manuscript. If necessary, provide both unit forms in the Introduction for clarity.*

AR: Thank you for your feedback. We have revised the units for $NH_3$ concentration in the Introduction to ensure consistency with the units used in the Results section. This modification should enhance the clarity and readability of the manuscript. The specific changes in the manuscript are as follows.

*Text in manuscript*

*1 Introduction*

....

Specifically, the annual average ammonia concentration across India is approximately $1.8$–$5.6 \times 10^{16}$ mol cm$^{-2}$, while in the Indo-Gangetic Plain (IGP) of India, the concentration is double that of

other regions, reaching a peak of $11.5 \times 10^{16} \text{ mol cm}^{-2}$ during the high season in July (Kuttippurath et al., 2020).

**RC:** *2) Both Metop-A and Metop-B data are used for 2019 but does not justify why both satellites are necessary instead of a single platform. Please clarify this choice in Section 2.1.*

AR: Thank you for your comment. The use of both Metop-A and Metop-B data for 2019 ensures data continuity and enhances the reliability of the observations. While a single satellite could provide sufficient data, using both platforms allows for redundancy, reducing the risk of data loss due to satellite malfunctions or other issues. Additionally, combining data from both satellites improves temporal and spatial coverage, resulting in more accurate and robust results. The specific changes in the manuscript are as follows.

*Text in manuscript*

*2.1 IASI satellite measurements*

…

Therefore, only the data from Metop-A and Metop-B within the 2019 time frame were used in this study. The use of both Metop-A and Metop-B data for 2019 ensures data continuity and enhances the reliability of the measurements. While a single satellite could provide sufficient data, using both platforms could improve temporal and spatial coverage, resulting in more accurate and robust results.

**RC:** *3) To ensure robustness, include a brief intercomparison of NH3 concentrations derived from Metop-A and Metop-B. Alternatively, compare the IASI version 4 products with prior studies.*

AR: Thank you for your suggestion. To ensure robustness, we have included a brief intercomparison of $NH_3$ concentrations derived from both Metop-A and Metop-B satellites. As shown in the Fig. S1, despite some small differences, the data from both satellites exhibit consistent spatial patterns and concentration levels. Furthermore, the data from the two satellites can complement each other, which enhances the reliability of the results. The specific changes in the manuscript are as follows.

*Text in manuscript*

*2.1 IASI satellite measurements*

The use of both Metop-A and Metop-B data for 2019 ensures data continuity and enhances the

reliability of the measurements. While a single satellite could provide sufficient data, using both platforms could improve temporal and spatial coverage, resulting in more accurate and robust results. To ensure robustness, we have also made a brief comparison of the $NH_3$ column concentrations obtained from both Metop-A and Metop-B satellites, as shown in the Fig. S1. Despite some small differences, the data from both satellites are generally consistent in terms of spatial patterns and concentration levels. Additionally, the data from the two satellites can complement each other, indicating good reliability of the results across both platforms.

a.1
IASI Metop-A $NH_3$ total column at 201901

a.2
IASI Metop-A $NH_3$ total column at 201904

a.3
IASI Metop-A $NH_3$ total column at 201907

b.1
IASI Metop-B $NH_3$ total column at 201901

b.2
IASI Metop-B $NH_3$ total column at 201904

b.3
IASI Metop-B $NH_3$ total column at 201907

**Figure S2. The distribution of IASI Metop-A (a) and Metop-B (b) $NH_3$ total column in 2019 January (a.1-b.1), April (a.2-b.2) and July (a.3-b.3).**

**RC:** *4) The exclusion of negative $NH_3$ total column values risks introducing a positive bias in gridded and monthly averages. Verify this by analyzing total columns in remote regions (e.g., oceans, deserts) where near-zero background concentrations are expected.*

AR: Thank you for your suggestion. We have re-compared the cases of excluding negative $NH_3$ total column values versus retaining them. As shown in the figure below, within our study region the effect on the final concentrations is minimal. However, we acknowledge that in remote regions (such as oceans and deserts) where near-zero background concentrations are expected, excluding negative values might introduce a positive bias. In future studies focusing on other regions, we will carefully re-evaluate and adjust our filtering criteria accordingly.

*Text in manuscript*

*2.1 IASI satellite measurements*

…

The assimilated observations for estimating the NH₃ emissions were the monthly IASI column concentration means over the 0.5 °× 0.625 °GEOS-Chem grid cell. These values were derived from the latest ANNI-NH₃-v4R-ERA5 product. Despite improvements in NH₃ column retrievals from satellite observations, there remains substantial variability in measurement uncertainty, ranging from 5% to over 1000% (Van Damme et al., 2014; Whitburn et al., 2016; Van Damme et al., 2017). Data selection was performed by excluding nighttime observations, irrational values (<0), and only using data with a cloud fraction < 0.1 (Van Damme et al., 2018) and skin temperature > 263 K (Van Damme et al., 2014) during the calculation of the monthly mean. While negative values are not necessarily incorrect, they are considered unrealistic in the context of NH3 concentrations. To improve the quality of the monthly average, we removed those negative values. Additionally, we have re-compared the cases of excluding negative NH₃ total column values versus retaining them. As shown in Fig. S2, the positive bias on the final concentrations within our study region is minimal.

[Figure]

*Figure S1. The distribution of IASI NH3 total column data in 2019. Panels (a.1)–(a.4) display the data after excluding negative values, while panels (b.1)–(b.4) display the data with negative values retained, corresponding to January, April, July, and November, respectively*

**RC: *5) GEOS-Chem emission inventory: The revised manuscript implies that only anthropogenic emissions (CEDS) are used to drive GEOS-Chem, omitting natural/biomass burning sources. If true: 1. The title should reflect this limitation (e.g., "anthropogenic NH₃ emissions" instead of general "NH₃ emissions"). 2. Justify the exclusion of natural/biomass burning emissions in Section 2.4, especially given the seasonal mismatch between modeled (summer) and biomass burning (spring) peaks. 3. Clarify why a higher-resolution inventory (e.g., MIXv2) was not adopted, as coarse resolutions may misrepresent localized emission hotspots. If not: add related info into the Sect. 2.4.***

AR: Thank you for your feedback. We actually considered both natural and biomass burning sources to drive the GEOS-Chem model, but in this study, we primarily estimated the anthropogenic ammonia emissions inventory (CEDS). We have added natural and biomass burning sources information in Section 2.4 to clarify this point. Additionally, we have revised the explanation of the seasonal variation in emissions to avoid confusion between anthropogenic sources and biomass burning sources. We now focus solely on the seasonal variation of anthropogenic sources. The title is now changed from "*South Asia ammonia emission inversion through assimilating IASI observations*" to "*South Asia anthropogenic ammonia emission inversion through assimilating IASI observations*"

We apologize for our previous inadequate response. Your recommendation is indeed useful. While high-resolution emission inversion is undoubtedly better, especially in capturing localized emission hotspots. Emission inversion for optimizing both the global and very local pattern also increases the computational load and requires more ensembles in our ensemble assimilation method. Otherwise, spurious correction introduced by the limited ensemble member will reduce the accuracy of the assimilation. Given our current limited computational resources, we chose not to adopt a higher-resolution inventory at this stage. However, we plan to consider using higher-resolution emission inventories, such as MIXv2, in future work to improve our results.

*Text in manuscript*
* * *
*2.4 GEOS-Chem model and emission inventory*

…

Specifically, these emissions stem predominantly from farmlands, including crops such as wheat, maize, and rice, as well as manure from livestock, including cattle, chicken, goats, and pigs (Liu et al., 2022). The CEDS emission estimates were coarse-grained into the model resolution $0.5° \times 0.625°$ before being utilized to drive the GEOS-Chem simulations. Examples of the CEDS emission over the South Asia are presented in Fig. 3, which plot the total $NH_3$ emission fluxes for January, April, July, November of the year 2019. Additionally, the model's biogenic emissions are based on the MEGAN2.1 (Model of Emissions of Gases and Aerosols from Nature) inventory (Guenther et al., 2012), while the biomass burning sources driving the model are based on the GFEDv4 (Version 4 of the Global Fire Emissions Database) inventory (Giglio et al., 2013). The use of IASI and CrIS observations, along with GEOS-Chem simulations, is outlined in Table 1.

…

| Country | Crop | Planting Period | Mid-Season | Harvest Period |
|---------|------|-----------------|------------|----------------|
| Bhutan | Corn | Feb–Mar | Apr–Jun | Jul–Sep |
| India | Corn (Kharif) | Mar–Jun | Jul–Aug | Sep–Oct |
| India | Cotton | Apr–Jul | Aug–Sep | Oct–Dec |
| India | Millet (Kharif, Pearl) | May–Jul | Aug | Sep–Nov |
| India | Peanut (Kharif) | May–Jul | Aug | Sep–Nov |
| India | Rice (Kharif) | May–Jul | Aug | Sep–Nov |
| India | Sorghum (Kharif) | May–Jul | Aug | Sep–Oct |
| India | Soybean | Jun–Jul | Aug | Sep–Oct |
| India | Sunflowerseed (Kharif) | Jun–Jul | Aug | Sep–Oct |
| Nepal | Millet | May–Jul | Aug | Sep–Nov |
| Nepal | Rice | May–Jul | Aug–Sep | Oct–Dec |
| Pakistan | Corn | May–Jul | Aug | Sep–Oct |
| Pakistan | Cotton | Mar–Jun | Jul–Aug | Sep–Nov |
| Pakistan | Millet | May–Jun | Jul | Aug–Sep |
| Pakistan | Peanut | Mar–Jun | Jul | Aug–Oct |
| Pakistan | Rice | May–Jul | Aug | Sep–Nov |
| Pakistan | Sorghum | Jun–Jul | Aug | Sep–Oct |
| Pakistan | Sunflowerseed | Jan–Feb | Mar–May | Jun |

**Table 2.** Crop calendars for selected Kharif crops in Bhutan, India, Nepal, and Pakistan from USDA.

**3.2 NH3 emissions analysis**

The substantial emissions in July, as indicated by the posterior inventory, can be attributed to the increased fertilizer application for crops during the summer season (Tanvir et al., 2019). As shown in Table 2, the sowing period for crops in South Asia is generally from May to July, with July being the peak growth period for crops, resulting in a large amount of fertilization, resulting in July surpassing May in emission intensity. From July to September, as rice and other crops progress through their growth stages, fertilizer application typically decreases, leading to a gradual reduction in $NH_3$ emissions. Additionally, temperatures decline from August to September Fig. S4 (b), reducing the volatilization rate of $NH_3$, thereby leading to a further decrease in emissions. This pattern occurs because $NH_3$ volatilization is strongly influenced by temperature (Fan et al., 2011).

**RC:** *6) Table 1: 'Data' -> 'Data and model'*

AR: Thank you for your feedback. Thank you for your suggestion. We have updated Table 1 to change "Data" to "Data and model" as per your recommendation.

*Table in manuscript*

*2.4 GEOS-Chem model and emission inventory*

| Data and Model | Period | Use |
|---|---|---|
| IASI v3 | 2015-2023 | Annual variation of $NH_3$ concentration |
| IASI v4 | entire 2019 | Inversion and Validation |
| Level 2 CrIS | entire 2019 | Independent validation |
| CPCB | entire 2019 | Independent validation |
| GEOS-Chem | entire 2019 | Similation |

**Table 1.** The use of observations and simulations

**RC:** *7) '3.2 Spatial and Seasonal variation of NH3 emission' -> 'NH3 emissions analysis'*

AR: Thank you for your suggestion. We have updated the section title from "3.2 Spatial and Seasonal variation of NH3 emission" to " Anthropogenic $NH_3$ emissions analysis" as recommended.

*Text in manuscript*

*3.2 Anthropogenic $NH_3$ emissions analysis*

**RC:** *8) The ms cites USDA-derived planting/harvesting times to explain NH3 seasonality but does not present this data in the main text or supplement.*

AR: Thank you for your comment. We have now included the relevant information regarding USDA-derived planting/harvesting times to explain $NH_3$ seasonality. The specific changes made to the manuscript are as follows:

*Text in manuscript*

*3.1.2 Seasonal and annual variation of NH3 concentration*

We have identified the planting and harvesting times of crops in the South Asia region fr om USDA(U.S.DEPARTMENT OF ARGRICULTURE, https://ipad.fas.usda.gov/rssiws/al/crop _calendar/sasia.aspx), as presented in Table 2.

| Country | Crop | Planting Period | Mid-Season | Harvest Period |
|---------|------|-----------------|------------|----------------|
| Bhutan | Corn | Feb–Mar | Apr–Jun | Jul–Sep |
| India | Corn (Kharif) | Mar–Jun | Jul–Aug | Sep–Oct |
| India | Cotton | Apr–Jul | Aug–Sep | Oct–Dec |
| India | Millet (Kharif, Pearl) | May–Jul | Aug | Sep–Nov |
| India | Peanut (Kharif) | May–Jul | Aug | Sep–Nov |
| India | Rice (Kharif) | May–Jul | Aug | Sep–Nov |
| India | Sorghum (Kharif) | May–Jul | Aug | Sep–Oct |
| India | Soybean | Jun–Jul | Aug | Sep–Oct |
| India | Sunflowerseed (Kharif) | Jun–Jul | Aug | Sep–Oct |
| Nepal | Millet | May–Jul | Aug | Sep–Nov |
| Nepal | Rice | May–Jul | Aug–Sep | Oct–Dec |
| Pakistan | Corn | May–Jul | Aug | Sep–Oct |
| Pakistan | Cotton | Mar–Jun | Jul–Aug | Sep–Nov |
| Pakistan | Millet | May–Jun | Jul | Aug–Sep |
| Pakistan | Peanut | Mar–Jun | Jul | Aug–Oct |
| Pakistan | Rice | May–Jul | Aug | Sep–Nov |
| Pakistan | Sorghum | Jun–Jul | Aug | Sep–Oct |
| Pakistan | Sunflowerseed | Jan–Feb | Mar–May | Jun |

**Table 2.** Crop calendars for selected Kharif crops in Bhutan, India, Nepal, and Pakistan from USDA.

**RC:** *9) The text attributes lower July NH3 concentrations (vs. May) to higher temperatures enhancing volatilization, which may be a counterintuitive claim.*

AR: We agree that attributing the lower July $NH_3$ concentrations (compared to May) to higher temperatures enhancing volatilization is not entirely appropriate. We have revised the explanation to better reflect the combined impact of high temperatures and increased precipitation on $NH_3$ concentrations. The specific changes made to the manuscript are as follows:

*Text in manuscript*

*3.1.2 Seasonal and annual variation of NH3 concentration*

*…*

The reasons for higher emissions in July but lower concentration levels compared to May could be attributed to meteorological factors. The monsoon season in South Asia results in increased wet deposition, and notably, 2019 experienced the most intense monsoon since 1994 (NASA, 2020). As shown in the Fig. S4 (a) and (b), precipitation and temperature in July are the highest of the year. High temperature could increase ammonia volatilization, leading to higher concentrations, while high precipitation increases the wet deposition of ammonia. However, the impact of temperature on concentration is secondary compared to the dramatic variations in precipitation. These combined

factors result in July having a smaller concentration peak compared to May, despite July being another peak month.

RC: *10) Add the number of your estimated budgets in the abstract and conclusion.*

AR: Thank you for your comment. I have already added the estimated budgets in both the abstract and conclusion sections of the document. The specific changes made to the manuscript are as follows:

*Text in manuscript*

**Abstract.** Ammonia has attracted significant attention due to its pivotal role in the ecosystem and its contribution to the formation of secondary aerosols. Developing an accurate ammonia emission inventory is crucial for simulating atmospheric ammonia levels and quantifying its impacts. However, current inventories are typically constructed in the bottom-up approach and are associated with substantial uncertainties. To address this issue, assimilating observations from satellite instruments for top-down emission inversion has emerged as an effective strategy for optimizing emission inventories. Despite the severity of ammonia pollution in South Asia, research in this context remains very limited. This study aims to estimate ammonia emissions in this region by integrating the prior emission inventory from the Community Emissions Data System (CEDS) and the columned ammonia concentration retrievals from the Infrared Atmospheric Sounder Interferometer (IASI). We employ a newly-developed four-dimensional ensemble variational (4DEnVar)-based emission inversion system to conduct the calculations, resulting in monthly ammonia emissions for 2019 at a resolution of $0.5° \times 0.625°$. The annual total estimate for the posterior emission inventory is 12.61 Tg, compared to the prior inventory's 13.32 Tg. Our simulations, driven by the posterior emission inventory, demonstrate superior performance compared to those driven by the prior emission inventory. This is validated through comparisons against the IASI observations, the independent column concentration measurements from the advanced satellite instrument Crosstrack Infrared Sounder (CrIS), and the ground concentration observations of ammonia and PM2.5. Additionally, the spatial and temporal characteristics of ammonia emissions in South Asia based on the posterior are analyzed. Notably, emissions there exhibit a "double-peak" seasonal profile, with the maximum in July and the secondary peak in May. This differs from the "double-peak" trend suggested by the CEDS prior inventory, which identifies the maximum column concentration in May and a second peak in September. The differences may be attributed to a more accurate representation of regional agricultural practices, such as the timing of fertilizer application and meteorological influences like precipitation and temperature.

*4 Summary and conclusion*

…

The spatial and temporal characteristics of $NH_3$ emissions over South Asia were then analyzed based

on the inversion. While the prior CEDS inventory generally captured the $NH_3$ emission hotspots, such as in Pakistan, North India, and Bengal, it failed to accurately represent the seasonal trend. Specifically, the prior inventory showed a "double-peak" pattern throughout the year, with peaks in May and September. In contrast, the posterior results revealed the correct seasonal pattern with the "double-peak" profile occurring in May and July. The posterior emission inventory's total annual estimate is 12.61 Tg, compared to the prior inventory's 13.32 Tg.